# The mitochondrial fusion protein OPA1 is dispensable in the liver and its absence induces mitohormesis to protect liver from drug-induced injury

Hakjoo Lee [1], Tae Jin Lee [2], Chad A. Galloway [3], Wenbo Zhi[2], Wei Xiao[2], Karen L. de Mesy Bentley [3], Ashok Sharma [2], Yong Teng [4], Hiromi Sesaki[5] & Yisang Yoon [1] ✉

Mitochondria are critical for metabolic homeostasis of the liver, and their dysfunction is a major cause of liver diseases. Optic atrophy 1 (OPA1) is a mitochondrial fusion protein with a role in cristae shaping. Disruption of OPA1 causes mitochondrial dysfunction. However, the role of OPA1 in liver function is poorly understood. In this study, we delete OPA1 in the fully developed liver of male mice. Unexpectedly, OPA1 liver knockout (LKO) mice are healthy with unaffected mitochondrial respiration, despite disrupted cristae morphology. OPA1 LKO induces a stress response that establishes a new homeostatic state for sustained liver function. Our data show that OPA1 is required for proper complex V assembly and that OPA1 LKO protects the liver from drug toxicity. Mechanistically, OPA1 LKO decreases toxic drug metabolism and confers resistance to the mitochondrial permeability transition. This study demonstrates that OPA1 is dispensable in the liver, and that the mitohormesis induced by OPA1 LKO prevents liver injury and contributes to liver resiliency.

The liver is responsible for the myriad of processes including the nutrient homeostasis and detoxification. Mitochondria in the liver are critical for nutrient metabolism, and therefore, dysfunction of mitochondria is one of the prominent causes for the loss of liver function and liver diseases[1–3]. Mitochondrial fission and fusion, the main processes of mitochondrial dynamics, determine mitochondrial shape, and have been recognized to be important for functional maintenance of mitochondria. Mitochondrial fusion may complement functional deficit and prolong mitochondrial life span, whereas fission may segregate dysfunctional mitochondria and aid autophagic removal of them (mitophagy)[4–8]. A subset of dynamin family of large GTPases are associated with mitochondria and mediate fission and fusion of mitochondrial membranes. Mitochondrial fission requires a cytosolic dynamin-related/like protein 1 (Drp1/DLP1) that is recruited to the mitochondrial outer surface upon fission signal[9,10]. Two isoforms of mitofusin, Mfn1 and Mfn2 are anchored at the mitochondrial outer membrane (OM) and mediate fusion of the OM, whereas the inner membrane (IM)-associated optic atrophy 1 (OPA1) fuses the IM upon fusion of the OM[9,10].

The functional significance of mitochondrial shape change has been demonstrated by the knockout (KO) mouse models of Drp1, Mfn1, Mfn2, and OPA1. Individual KOs of these mitochondrial dynamics genes in mice cause embryonic lethality, indicating the critical roles of these proteins in animal development[11–14]. Furthermore, tissue-

[1]Department of Physiology, Medical College of Georgia, Augusta University, Augusta, GA 30912, USA. [2]Center for Biotechnology and Genomic Medicine, Medical College of Georgia, Augusta University, Augusta, GA 30912, USA. [3]Department of Pathology and Laboratory Medicine, and Center for Advanced Research Technologies, University of Rochester Medical Center, Rochester, NY 14642, USA. [4]Department of Hematology and Medical Oncology, Emory University School of Medicine, Atlanta, GA 30322, USA. [5]Department of Cell Biology, Johns Hopkins University School of Medicine, Baltimore, MD 21205, USA. ✉ e-mail: yyoon@augusta.edu

specific KOs in energy-demanding organs such as brain and heart result in organ dysfunction and animal death[11,12,15], indicating that mitochondrial fission and fusion are critical for functional maintenance of mitochondria.

OPA1, the known IM fusion protein, has an additional function in maintaining cristae structure, thus having a more direct role in maintaining mitochondrial function[16–19]. OPA1 was proposed to maintain cristae junction and cristae tightness by making cross-bridges at the junction and along the cristae membrane[16,19]. Global knockout (KO) of OPA1 in mice cause embryonic lethality, indicating its critical roles in animal development[14]. Furthermore, OPA1 deletions in mitochondria-dependent organs as well as in endothelial cells cause respiration defect and are lethal, demonstrating the critical role of OPA1 for maintaining mitochondrial function to support cell viability[20–23]. Consistent with this notion, it has been shown that OPA1 overexpression is beneficial in experimental models of certain diseases including apoptotic liver injury[24,25]. On the other hand, OPA1 depletion in the mouse liver was shown to rather decrease the non-alcoholic fatty liver disease (NAFLD) and nonalcoholic steatohepatitis (NASH) pathologies[26,27]. These opposing observations raise questions as to the OPA1 function in the liver.

Liver mitochondria are frequently exposed to harmful materials such as reactive species and adducts from food and drug metabolism. Therefore, maintaining functional mitochondria under stressful environment will be a key to the long-term preservation of liver function. Mitohormesis is a re-equilibration process of mitochondrial and cellular functions in response to various forms of mitochondrial stresses[28]. Mitohormesis can also protect cells from future insults, and thus, it can be an important factor for protecting the liver. Drug-induced liver injury (DILI) occurs through mitochondrial dysfunction and is a common cause of acute and chronic liver diseases. Most of the drugs are metabolized in the liver through the cytochrome P450 (CYP) system[29,30]. Reactive intermediates and adducts from the CYP-drug reactions damage cellular components including mitochondria, which is a main cause of DILI[31].

In the current study, we found an unexpected beneficial effect of deleting OPA1 in the liver. Given the role of OPA1 in structural and functional maintenance of mitochondria and liver being the central metabolic organ, we expected a severe, if not lethal, phenotype of OPA1-liver KO (LKO) mice. However, OPA1-LKO mice were healthy and showed no ill phenotypes with normal mitochondrial respiration. Our analyses showed that OPA1 LKO induced alterations in liver and mitochondrial proteomes through a stress response. We identified a connection between OPA1 and proper assembly of the respiratory complex V, specifically $F_O$. Our data indicate that OPA1-KO induces the accumulation of unassembled complex V sub-complexes, suggesting a mitochondrial proteostatic stress in OPA1-KO liver. Importantly, OPA1-KO livers were protected from DILI, indicating that OPA1 KO induces mitohormesis. Mechanistically, we found that OPA1 KO in the liver decreases the toxic drug metabolism as well as the sensitivity to mitochondrial permeability transition (MPT). Our data indicate that the liver mitohormesis contributes to the liver resiliency.

## Results
### Deletion of OPA1 in the liver induces a halt in weight gain with no ill health effect

We introduced adeno-associated virus (AAV) that carries hepatocyte-specific Cre (AAV8-TBG-Cre) to the 8-week-old OPA1-floxed mice to generate OPA1-LKO mice. Cre was expressed for 8-12 weeks to allow sufficient time for OPA1 deletion to reveal functional consequences. AAV8-TBG-GFP was used for generating control mice. OPA1 was specifically removed from the liver, as unchanged levels of OPA1 were found in other tissues (Supplementary Fig. S1a). Despite the efficient KO of OPA1 in the liver, OPA1-liver KO (LKO)

mice had no apparent health problem, exhibiting sleek appearance and normal behavior.

Weekly measurements of body weights revealed that OPA1-LKO mice stopped gaining weight 2–3 weeks after AAV-Cre administration, showing a statistically significant difference from floxed control mice at 4 weeks post AAV ($p = 0.0005$) (Fig. 1a, b). While the control mice continued gaining weight, the body weight of OPA1-LKO mice increased in the first 2 weeks and then decreased to the initial weight before it was stabilized at 5 weeks post AAV (Fig. 1b). The liver weight normalized against body weight was similar to control after 8–12 weeks of AAV injection (Supplementary Fig. S1b). As there is a significant weight reduction in OPA1-LKO mice, we compared body fat content by computerized tomography (CT) scan (Supplementary Fig. S1c). While the size difference was noticeable, total fat volume showed no difference between control and OPA1-LKO mice (Supplementary Fig. S1d). We measured blood glucose levels and found that there were no differences in both non-fasted and fasted blood glucose (Fig. 1c). However, glucose tolerance test (GTT) indicates that OPA1 LKO results in faster blood glucose clearance, consistent with the reduced body weight (Fig. 1d). Evaluation of metabolic parameters indicated that OPA1-LKO mice show significantly higher $VO_2$ and $VCO_2$ (Fig. 1e, f). Calculated respiratory exchange ratio (RER) was also higher in OPA1-LKO mice, suggesting a decrease in fat usage by OPA1 LKO (see below) (Fig. 1g). OPA1-LKO mice showed a decrease in energy expenditure (Fig. 1h), likely reflecting their smaller body size. Interestingly, OPA1-LKO mice consumed food in both light and dark cycles whereas control mice showed a typical nocturnal feeding (Fig. 1i), which was also evident in RER and other parameters. Hence, a small increase in XYZ movement in light cycle was observed in OPA1-LKO mice (Fig. 1j). These changes in metabolic parameters of OPA1-LKO mice indicate an overall increase in metabolism with a change in feeding behavior.

Histology of OPA1-KO livers showed no sign of necrosis and injury (Fig. 2a). OPA1-KO livers showed no pathology, but had enlarged hepatocytes, similar to previous observation with OPA1-KO by Alb-Cre[26]. Quantification by the number of cells per unit area showed a smaller number of cells in OPA1-KO liver, indicating cell enlargement (Fig. 2b). To test the effect of OPA1 LKO on liver function, we tested serum levels of alanine aminotransferase (ALT) and aspartate aminotransferase (AST). We found that OPA1 deletion for 8-12 weeks increased both ALT and AST levels (Fig. 2c, d). The mean ALT of control mice was 22.5 U/L whereas that of OPA1-LKO mice 72.5 U/L. Similarly, the mean AST of floxed and OPA1-LKO mice were 50 and 134 U/L, respectively. Although these increases were statistically significant, both these ALT and AST values are within the reference range of C57BL/6 (22–133 U/L for ALT and 46–221 U/L for AST)[32,33]. It is likely that the increases of serum ALT and AST are a reflection of the increased hepatic expression of them in OPA1-LKO mice (see below), and may not be an increase of liver injury. Consistent with this notion, the cell injury marker serum lactate dehydrogenase (LDH) was not different in floxed and OPA1-KO mice (Fig. 2e). Immunoblotting of liver lysates for caspase 3 showed no increase of the cleaved active form in OPA1-KO liver, indicating no apoptotic cell death by OPA1 KO (Fig. 2f). Quantification indicates rather decreased caspase cleavage in OPA1-KO liver (Fig. 2g). Accordingly, there was no 89-kDa PARP-1 fragment generated by active caspase 3 in OPA1-KO liver (Fig. 2f). Two PARP-1 fragments in the 55–75 kDa range were observed, presumably produced by cathepsins[34]; however, one of these was rather decreased in OPA1-KO livers (Fig. 2f). In addition, OPA1-KO livers showed a small decrease in the LC3-II/I ratio compared with control livers, suggesting a decrease in autophagy (Fig. 2f, g). This series of observations indicate that OPA1 KO in fully developed mouse liver causes no ill effect on hepatic viability, liver function and animal health while restricting age-dependent weight gain, improving glucose tolerance, and enhancing whole body metabolism.

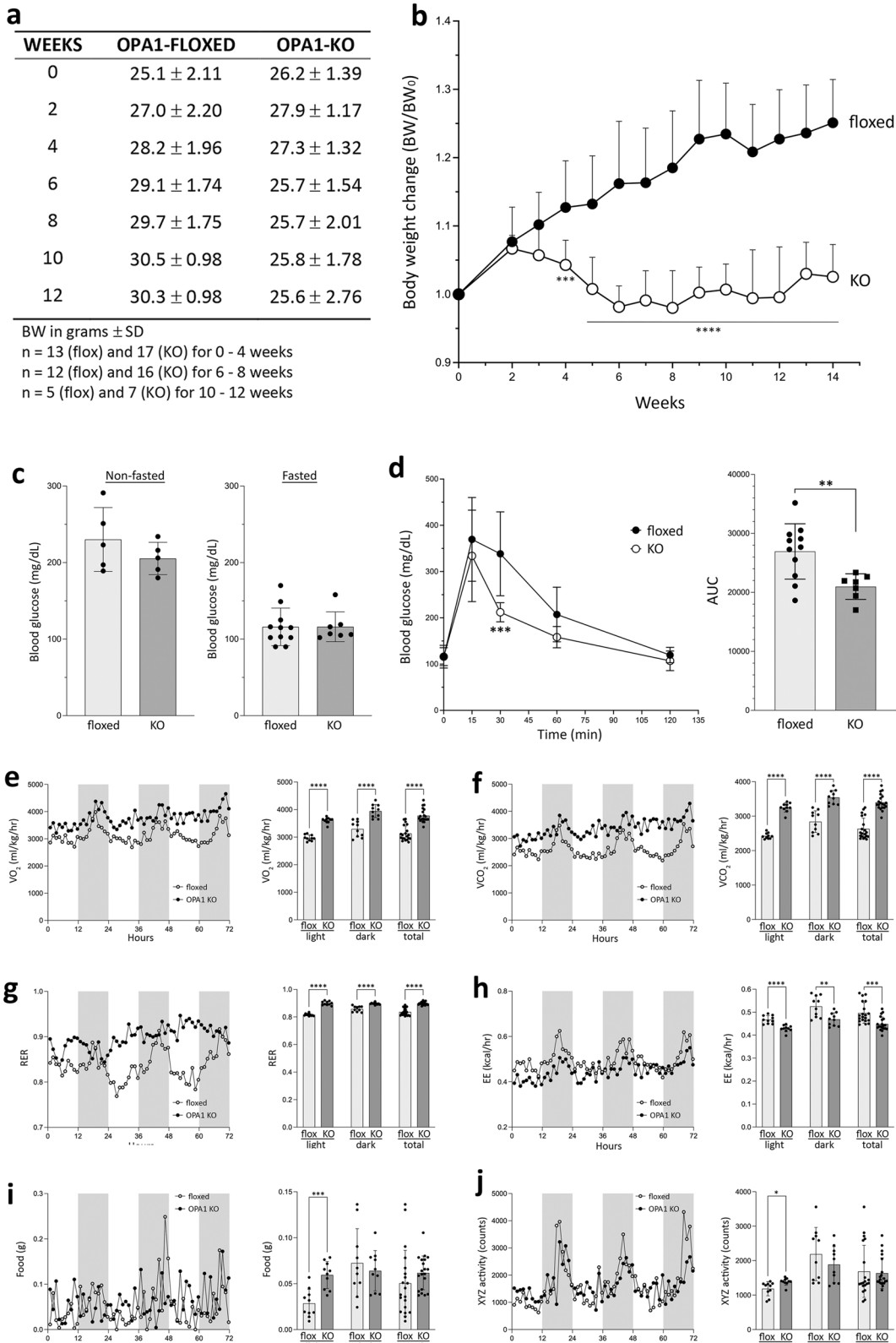

| WEEKS | OPA1-FLOXED | OPA1-KO |
|---|---|---|
| 0 | $25.1 \pm 2.11$ | $26.2 \pm 1.39$ |
| 2 | $27.0 \pm 2.20$ | $27.9 \pm 1.17$ |
| 4 | $28.2 \pm 1.96$ | $27.3 \pm 1.32$ |
| 6 | $29.1 \pm 1.74$ | $25.7 \pm 1.54$ |
| 8 | $29.7 \pm 1.75$ | $25.7 \pm 2.01$ |
| 10 | $30.5 \pm 0.98$ | $25.8 \pm 1.78$ |
| 12 | $30.3 \pm 0.98$ | $25.6 \pm 2.76$ |

BW in grams $\pm$ SD
n = 13 (flox) and 17 (KO) for 0 - 4 weeks
n = 12 (flox) and 16 (KO) for 6 - 8 weeks
n = 5 (flox) and 7 (KO) for 10 - 12 weeks

## Mitochondria in OPA1-KO liver have altered cristae structure but maintain normal respiration capacity

Electron microscopy (EM) of OPA1-KO liver showed mitochondria with decreased number of cristae (Fig. 3b). These mitochondria often bore very short cristae, apparently growing out of the inner boundary membrane (IBM) (Fig. 3c-f), which we referred to as primordial cristae[17]. Cristae in OPA1-KO mitochondria were often detached from

cristae junction. Some of detached cristae maintained long tubular form, while others formed short tubules and vesicles (Fig. 3c–f). Although OPA1 has been proposed to form intracristal cross-bridges along the cristae membrane and at the cristae junction to maintain cristae width[16,19], we observed that long tubular cristae still maintained narrow cristae width in OPA1-KO mitochondria. Some mitochondria in OPA1-KO liver had septum-like long cristae that completely or partially

**Fig. 1 | OPA1 KO in the liver induces a halt in weight gain with enhanced whole body metabolism. a** Body weights (BW) of floxed and OPA1-LKO mice after AAV-Cre injection. The week 0 is the day of injection. **b** Weekly changes of body weights normalized to the initial weight are plotted. $n = 13$ (flox) and 17 (KO) animals for 0–5 weeks; $n = 12$ (flox) and 16 (KO) animals for 6–8 weeks; $n = 6$ (flox) and 8 (KO) animals for week 9; $n = 5$ (flox) and 7 (KO) animals for 10–14 weeks. Data are presented as mean values +/- SD. Two-way ANOVA with Sidak's multiple comparisons. ***$p = 0.0005$, ****$p < 0.0001$. **c** Blood glucose measurements in non-fasted and fasted conditions. $n = 5$ animals per group for nonfasted; $n = 10$ (flox) and 7 (KO) animals for fasted. Data are presented as mean values +/- SD. Unpaired t test. **d** Glucose tolerance test (GTT) showing faster glucose clearance in OPA1-LKO mice. $n = 11$ (flox) and 7 (KO) animals. Data are presented as mean values +/- SD. Two-way ANOVA with Sidak's multiple comparisons. ***$p = 0.0001$. Area under curve (AUC) shows an improved glucose clearance by OPA1 KO. Data are presented as mean values +/- SD. Unpaired t test with Welch's correction. **$p = 0.0024$. **e–j** Assessment of metabolic parameters. $n = 4$ for floxed and OPA1-LKO mice. Hourly average is plotted for 72 hours. Shaded periods denote the dark cycle. Error bars are omitted. Bar graphs are the 3-day hourly average for light and dark cycles as well as total. $n = 4$ animals per group over 10 measures for light and dark cycles and 20 measures for total. Data are presented as mean values +/- SD. Multiple unpaired t test. The p values for light, dark, and total, respectively, are: (**e**) VO$_2$, $p < 0.000001$, $p < 0.000034$, and $p < 0.000001$; (**f**) VCO$_2$, $p < 0.000001$, $p = 0.000013$, and $p < 0.000001$; (**g**) respiratory exchange ratio; $p < 0.000001$, $p = 0.000019$, and $p < 0.00001$; (**h**) energy expenditure, $p = 0.000032$, $p = 0.004018$, and $p = 0.000340$; (**i**) food consumption, $p = 0.000209$, $p = 0.546617$, and $p = 0.217867$; (**j**) total activity, $p = 0.011132$, $p = 0.317263$, and $p = 0.803317$. Source data are provided as a Source Data file.

traversed and divided the matrix (Fig. 3e, arrow). This might have resulted from the absence of IM fusion due to OPA1-KO after fusion of the outer membranes. Mitochondria in OPA1-KO fibroblasts were shown to have drastically decreased matrix electron density, a trait of mitochondrial dysfunction[17]. However, most mitochondria in OPA1-KO liver still maintained matrix electron density despite altered cristae, suggesting mitochondrial function in OPA1-KO liver may still be maintained.

Oxygen consumption rate (OCR) analyses indicated that OPA1 KO did not cause significant changes in basal, state 3, leak, and maximum OCRs (Fig. 3g). No significant differences were observed in ATP-linked and reserve capacity OCRs as well as respiratory control ratio (RCR) (Fig. 3h–j). In addition, there was no difference in total ATP levels in control and OPA1-KO livers (Fig. 3k). These results in liver are surprising to us because OPA1 KO or silencing in other cells and tissues have been shown to decrease respiratory capacity of mitochondria[20–23,35,36]. It is also surprising that the structural change of cristae did not affect respiration. It is possible that the mitochondrial isolation process might have selected healthy mitochondria from OPA1-KO liver. However, EM of isolated mitochondria showed that mitochondria isolated from OPA1-KO liver had altered internal structure, indicating no selection for normal mitochondria. Control mitochondria generally showed wider cristae compared with those in situ, probably due to an osmotic change during isolation[37,38] (Fig. 3l). Unlike control mitochondria, OPA1-KO mitochondria had numerous, but markedly ballooned cristae attached to the IBM that were likely primordial cristae in situ (Fig. 3m). These balloons were both small and large likely originated from very short and more developed primordial cristae, respectively. Despite the absence of the OPA1 protein, narrow cristae junction was maintained in ballooned cristae (arrowheads, Fig. 3n). A few large swollen vacuole-like cristae were also found in OPA1-KO mitochondria, but these could still be attached to the cristae junction or continuous with the IBM (arrows, Fig. 3m, o). These EM observations support the notion that OPA1 plays a role in maintaining cristae structure, but not cristae junction. Interestingly, whereas swollen and ballooned cristae of OPA1-KO mitochondria have smoothly curved edges, cristae edges of control mitochondria are straight and angular, suggesting a presence of structural constraint preventing smooth ballooning. OPA1 supposedly forms intracristal cross-bridges, and the finite length of the OPA1 cross-bridge would define the narrowness of cristae observed in tissues. However, the cristae width of isolated mitochondria with normal OPA1 is much wider than that in situ, suggesting that the OPA1 cross-bridge may become structurally unstable during mitochondrial isolation. Alternatively, because narrow cristae width is still maintained in OPA1-KO mitochondria in situ (Fig. 2b-f), it is possible that factors other than OPA1 maintain cristae narrowness. Our observations showed that the OXPHOS activity is maintained in OPA1-KO liver mitochondria despite the cristae alteration. Respiratory complexes are situated within the cristae membrane. Despite morphological alterations of cristae, there would be sufficient cristae membranes in OPA1-KO mitochondria where respiratory complexes formed and supported respiration.

## OPA1 KO alters liver proteome to preserve liver function

To understand the nature of the absence of functional defect in OPA1-KO liver, we performed the comparative proteomics of liver lysate and isolated liver mitochondria from control and OPA1-LKO mice. The protein heatmaps for >2-fold changes in peptide spectrum match (PSM) scores indicate that OPA1 LKO induces significant alterations in both mitochondrial and liver proteomes (Fig. 4a). Notable proteins increased in OPA1 KO include Lon protease LonP1 (LONM) and a mitochondrial stress-70 chaperone (GRP75), which are known to be induced under stress[39] (Fig. 4b). The level of ALT (ALAT2) was also increased in OPA1-KO liver approximately by 8-fold (Fig. 4b). A small but significant increase of AST was also observed in OPA1 KO liver ($\log_{10}P = 5.6$). These hepatic increases likely accounts for the observed increase in serum ALT and AST (Fig. 2). Ingenuity Pathway Analysis (IPA, Qiagen) showed that sirtuin signaling and the TCA cycle were activated, suggesting the effort to support mitochondrial function (Supplementary Fig. S2a). The most repressed canonical pathway is EIF2 signaling, which suggests the activation of the integrated stress response (ISR) by OPA1 KO. Also decreased were LXR/RXR signaling, lipid degradation, oxidative phosphorylation, fatty acid oxidation, and multiple xenobiotic metabolisms. A decrease in fatty acid oxidation by OPA1 LKO might be manifested in an increase of RER (Fig. 1). These predicted decreases suggest general attenuation of liver activity by OPA1 KO. Upstream regulator prediction by IPA shows activations of growth factor/hormone signaling (Rictor, CGA, PTEN, InsR, etc.) along with PGC1α (PPARGC1A) for metabolic homeostasis and mitochondrial biogenesis, supporting functional maintenance of OPA1-KO liver (Supplementary Fig. S2b). The predicted increase in insulin receptor signaling in OPA1-LKO mice is consistent with their augmented insulin sensitivity evaluated by GTT (Fig. 1d).

A decrease of the EIF2 signaling in OPA1-KO liver suggests an activation of the ISR that is a general stress response[40,41]. The central regulator of ISR is the eukaryotic initiation factor 2 alpha (eIF2α) and its phosphorylation is an indicator of the ISR induction. Indeed, we observed a strong increase of eIF2α phosphorylation in OPA1-KO liver (Fig. 4c, d). Hepatic fibroblast growth factor 21 (FGF21) is the major ISR target protein and is known as a mitokine that is a downstream effector of mitochondrial stress[42–45]. Immunoblotting showed a significant increase of the FGF21 level in OPA1-KO (Fig. 4c, d), indicating that ISR-induced FGF21 increase may support functional maintenance of mitochondria. In addition, FGF21 is known to induce PGC1α expression[46–48]. We found a significant increase of PGC1α along with Tom20 and cytochrome $c$ in OPA1-KO livers, consistent with the increased mitochondrial biogenesis (Fig. 4c, d). As circulating FGF21 is mainly liver-derived[49], the serum level of FGF21 was also increased (Supplementary Fig. S2c). Systemic administration of FGF21 was shown to rectify insulin resistance and

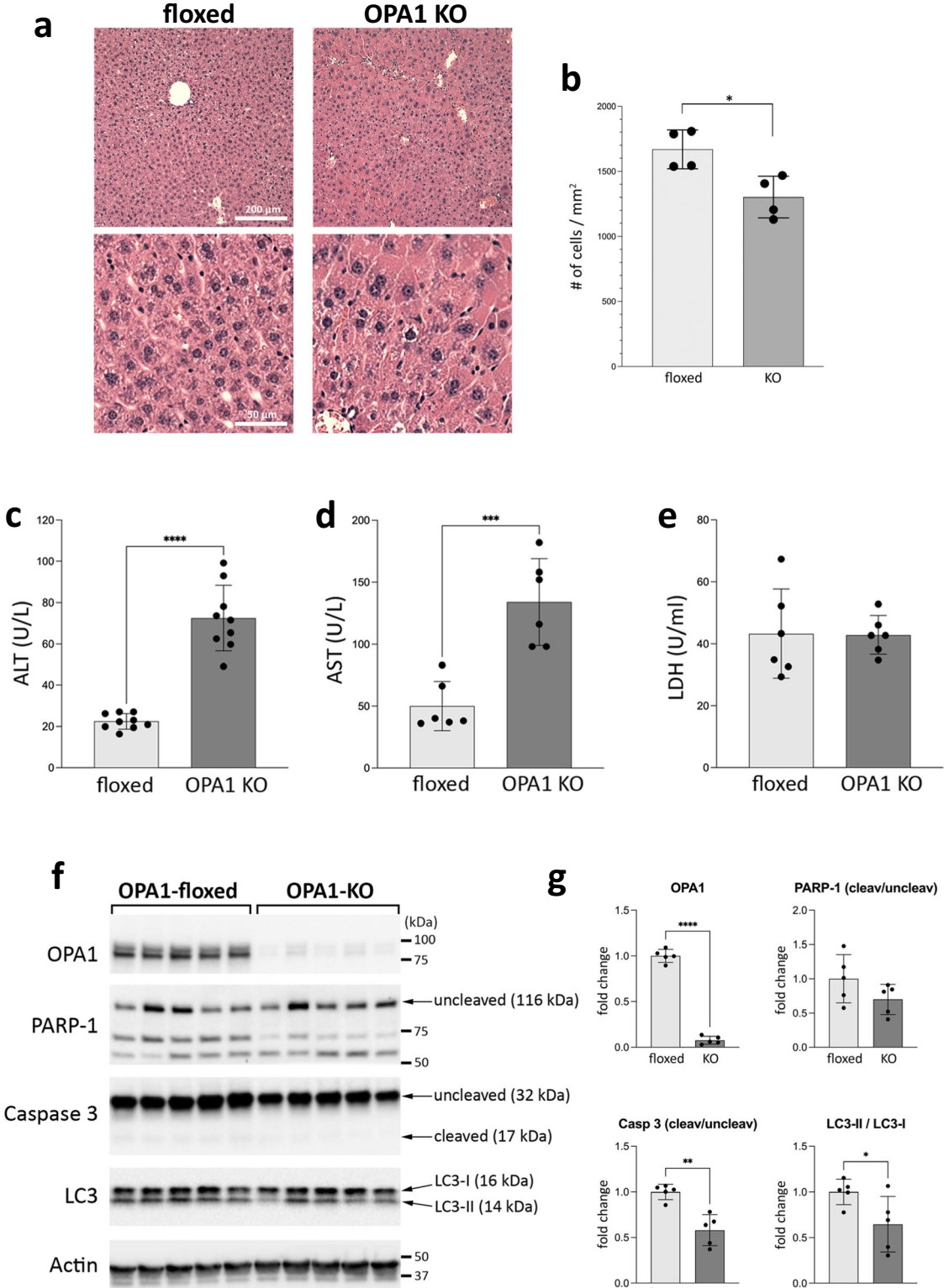

**Fig. 2 | OPA1 LKO causes no liver injury. a** H&E staining of liver sections. 4x enlarged images are also shown. **b** The number of cells in 1 mm² from H&E-stained liver section. Smaller cell numbers in a unit area in OPA1-KO liver indicate larger size cells. $n = 4$ animals per group. Data are presented as mean values +/- SD. Unpaired t test. $p = 0.0154$. **c** Serum ALT measurements. $n = 9$ animals per group. Data are presented as mean values +/- SD. Unpaired t test. $p < 0.0001$. **d** Serum AST measurements. $n = 6$ animals per group. Data are presented as mean values +/- SD. Unpaired t test. $P = 0.0005$. **e** Serum LDH measurements. $n = 6$ animals per group. Data are presented as mean values +/- SD. Unpaired t test. **f** Immunoblots of floxed and OPA1-KO livers for markers of apoptosis and autophagy. **g** Quantification of (**f**). Significantly decreased caspase 3 cleavage and LC3-II/LC3-I ratio in OPA1-KO liver. $n = 5$ animals per group. Data are presented as mean values +/- SD. Unpaired t test. ****$p < 0.0001$; **$p = 0.0011$; *$p = 0.0463$. Source data are provided as a Source Data file.

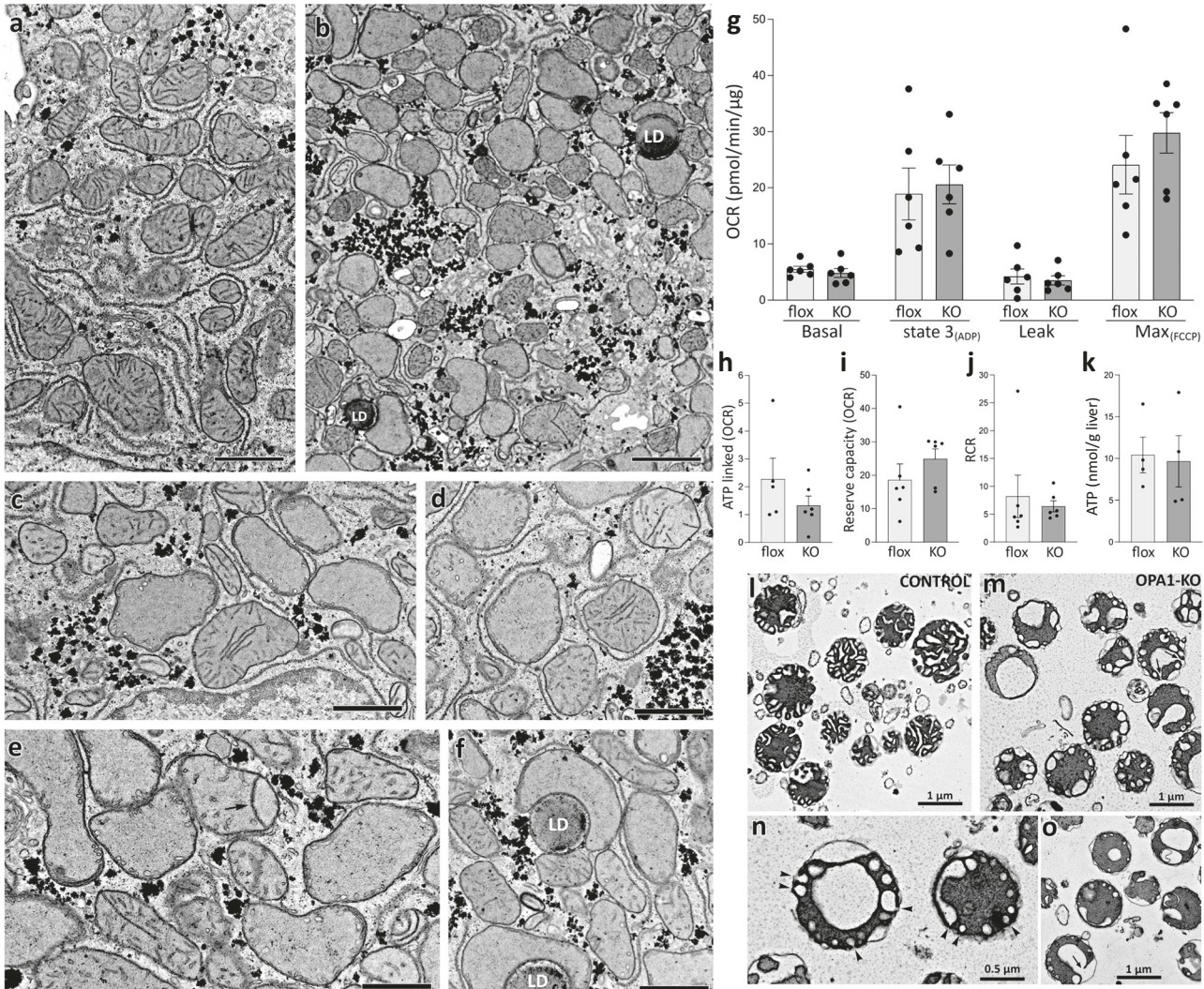

**Fig. 3 | Mitochondria in OPA1-KO liver have disrupted cristae, but maintain normal respiratory capacity.** Electron micrographs of thin sections from floxed (**a**) and OPA1-KO livers (**b**–**f**). Many OPA1-KO mitochondria lack normal cristae, but maintain their electron density (**b**). A number of mitochondria in OPA1-KO liver have primordial cristae or junction-less cristae (**c**–**f**). LD: lipid droplet. Scale bars in (**a**) and (**b**) are 2 μm and (**c**–**f**) are 1 μm. One liver from a floxed animal and two livers from KO animals were processed. The results were consistent. **g** Respiration analyses. Basal (in glutamate/malate), state 3, leak, and maximum OCRs were measured by sequential additions of ADP, oligomycin, and FCCP, respectively. $n = 6$ animals per group. Data are presented as mean values +/- SEM. Unpaired t test. **h**–**k** No significant differences in ATP-linked ( = Basal−Leak), reserve capacity ( = Max−Basal) and respiratory control ratio (state 3/state 4o) as well as liver ATP content. $n = 6$ animals per group. Data are presented as mean values +/- SEM. Unpaired t test. Source data are provided as a Source Data file. **l** EM of isolated mitochondria from control (floxed) liver. Note wider cristae compared with those in mitochondria in situ. **m**–**o** EM of isolated mitochondria from OPA1-KO liver. Cristae ballooning is evident. These cristae are mostly attached to the inner boundary membrane by junctions (arrows and arrowheads). One mitochondrial preparation each from floxed and KO animals were processed.

obesity[50–52]. Thus, the halt in age-dependent weight gain and improved glucose tolerance of OPA1-LKO mice are possibly the systemic effect of the increased level of circulating FGF21. Overall, observed changes in the liver proteome suggest that OPA1 KO induces a large-scale change in protein expression and liver metabolism through the ISR, supporting liver function.

## OPA1 KO changes respiratory complexes and supercomplexes

Our proteomic analyses show that OPA1 KO significantly decreased multiple subunits of complex I and IV, whereas increased subsets of complex II and V subunits (Fig. 5a, b). The cytochrome *c* level also increased significantly in OPA1 KO. Interestingly, marked increases in assembly factors of both complex I (Q59J78) and IV (Q921H9) were observed in OPA1-KO mitochondria, suggesting a compensatory effect caused by decreased levels of the respective complexes. Immunoblotting confirmed the significant decreases in the subunits of complexes I and IV, and an increase in those of complexes II and V in OPA1-KO livers (Supplementary Fig. S3a, b).

We used blue-native gel electrophoresis (BNGE) to define the effect of the identified changes in respiratory subunit levels on complex and supercomplex assembly. Most of the complex I was found as $[I + III_2]$ and $[I + III_2 + IV]$ supercomplexes, and OPA1 KO significantly decreased their levels (Fig. 5c, g). OPA1 KO increased the complex II level (Fig. 5e, g), which is likely due to enhancement of TCA cycle for supporting anaplerotic reactions, as complex II is the TCA cycle enzyme succinate dehydrogenase. Complex III did not show a significant change although a slight increase in the $[III_2]$ dimer was observed (Fig. 5d, g), presumably due to the decreases in the $[I + III_2]$ and $[I + III_2 + IV]$ formations. Significant decreases of complex IV and complex IV-containing supercomplexes were observed in OPA1-KO mitochondria with a small accumulation of a complex IV subassembly (Fig. 5f, g).

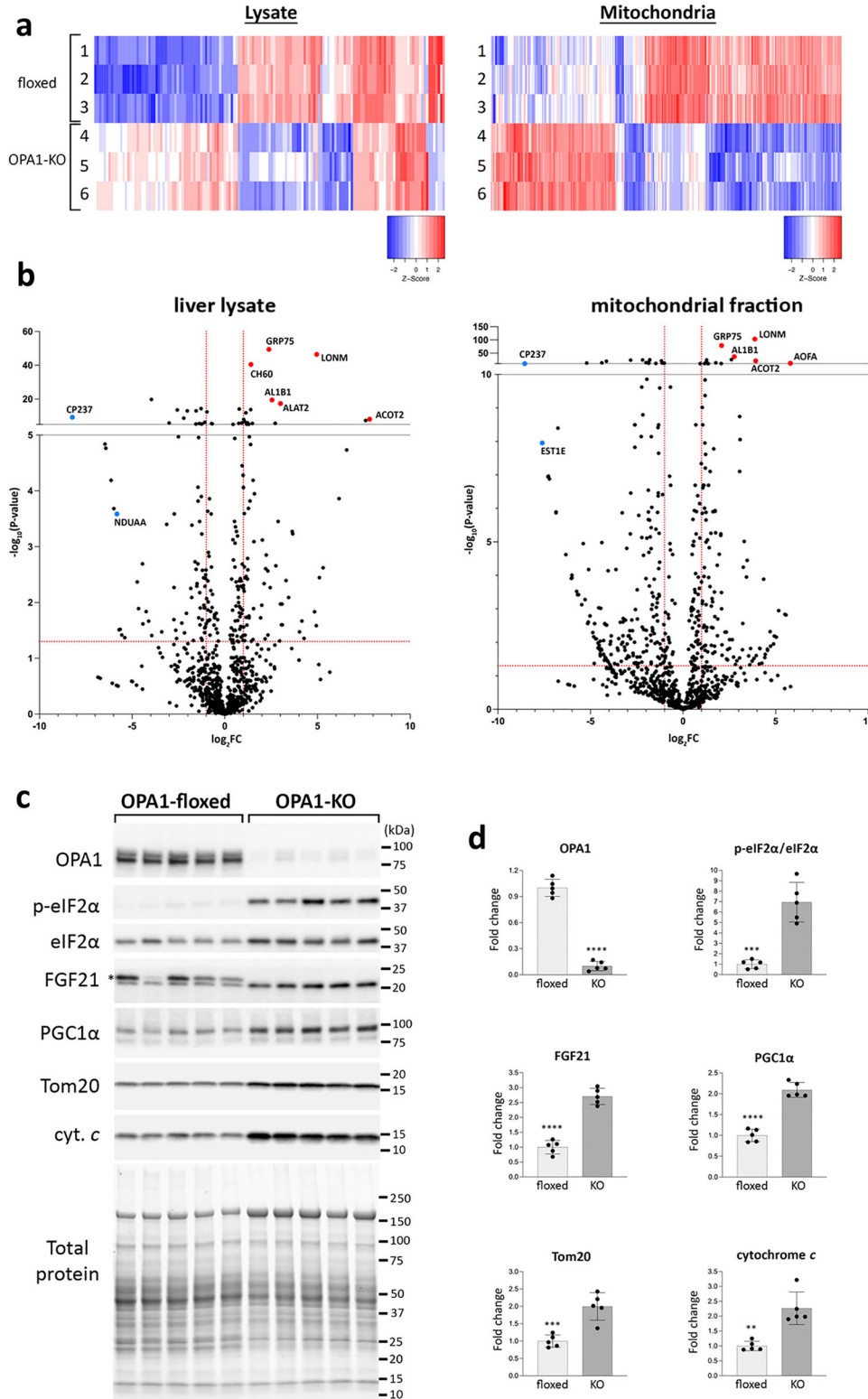

**Fig. 4 | OPA1 LKO alters the liver proteome by the ISR. a** The protein heatmaps for >2-fold changes in PSM scores in the lysate and mitochondrial fraction from floxed (1–3) and OPA1-KO (4–6) livers. Protein IDs are omitted. **b** Volcano plots of proteins identified by mass spectrometry. Red dashed lines are the cut-offs for significance: FC > 2 in X axis, and $p < 0.05$ in Y axis. GRP75, mitochondrial stress-70 protein; LONM, mitochondrial Lon protease; CH60, mitochondrial 60 kDa heat shock protein; AL1B1, mitochondrial aldehyde dehydrogenase X; ALAT2, alanine aminotransferase 2; CP237, cytochrome P450 2C37; ACOT2, mitochondrial

acyl-coenzyme A thioesterase 2; AOFA, amine oxidase [flavin-containing] A; NDUAA, mitochondrial NADH dehydrogenase [ubiquinone] 1 alpha subcomplex subunit 10; EST1E, carboxylesterase 1E. **c** Immunoblotting of liver lysates from control (floxed) and OPA1-KO mice. Asterisk denotes a nonspecific band. **d** Quantification of immunoblots. $n = 5$ animals per group. Data are presented as mean values +/- SD. Unpaired t test. ****$p < 0.0001$; ***$p = 0.0001$ (p-eIF2α) and $p = 0.0009$ (Tom20); **$p = 0.0011$. Source data are provided as a Source Data file.

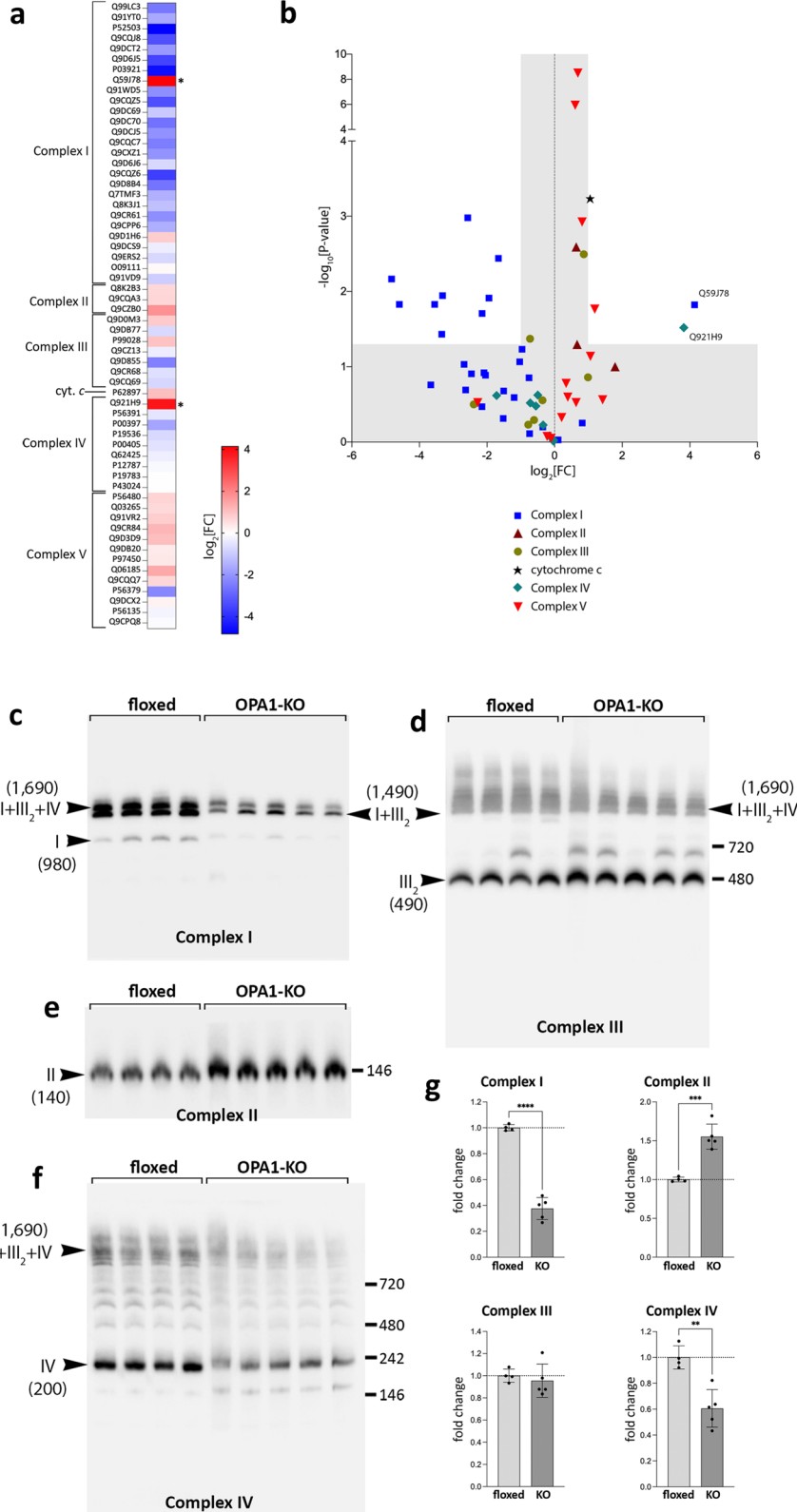

**Fig. 5 | OPA1 KO alters the levels of respiratory complex subunits and formation of supercomplexes. a** The heatmap of respiratory complex subunits. Uniprot IDs are shown. **b** Volcano plot of respiratory complex subunits. Proteins with a statistically significant change are in the unshaded area. Q59J78, NADH dehydrogenase [ubiquinone] 1 alpha subcomplex assembly factor 2; Q921H9, cytochrome c oxidase assembly factor 7. **c**–**f** Immunoblots of the BNGEs for respiratory complexes. OPA1 KO decreases complex I (**c**) and complex IV (**f**), and the complex

I-containing supercomplexes [I + III$_2$] and [I + III$_2$ + IV] (**c**, **d**, and **f**). Increased complex II levels in OPA1 KO (**e**). The numbers in the parentheses are the molecular weights in kDa. Each lane represents mitochondrial extract from different animals. **g** Quantification of the total levels of complexes I–IV. $n = 4$ (flox) and 5 (KO) animals per group. Data are presented as mean values +/- SD. Unpaired t test. ****$p < 0.0001$; ***$p = 0.0003$; **$p = 0.0022$. Source data are provided as a Source Data file.

It is interesting to observe the ~700-kDa complex that is present in some and absent in others in the complex III blot (Fig. 5d). This size is similar to that of the $[III_2 + IV]$ supercomplex (690 kDa). The protein Cox7a21, also known as SCAF1 (supercomplex assembly factor 1), was shown to be necessary for the formation of $[III_2 + IV]$[53]. C57BL/6 mice were found to contain a 2 amino acid shorter version of the SCAF1 protein (111 amino acids vs. WT 113 amino acids), which makes them defective for the $[III_2 + IV]$ formation[53]. OPA1-floxed mice were originally in a C57BL/6-129/SvEv mixed background[54], and the varied presence of the 700-kDa complex among OPA1-floxed and KO mice (Fig. 5d) suggests that it might be due to differentially segregated *SCAF1* alleles. Although structural and bioenergetic roles of SCAF1 are debatable[55,56], our PCR analyses showed the exact match of the presence of the 700-kDa band with that of $SCAF1_{113AA}$, indicating that the 700 kDa band is likely the $[III_2 + IV]$ supercomplex (Supplementary Fig. S3c, d). The PCR data show that the mice forming the $[III_2 + IV]$ were $SCAF1_{111AA}$ / $SCAF1_{113AA}$ heterozygotes, whereas the rest were $SCAF1_{111AA}$ homozygotes (Supplementary Fig. S3c, d). These data support the structural role of SCAF1 in forming the $[III_2 + IV]$ supercomplex, while we have not observed any correlation between SCAF1 and functional aspect of mitochondria. Irrespective of the *SCAF1* allele, OPA1-LKO mice consistently showed halted weight gain, ISR induction, and changes in respiratory complexes.

Overall, these changes in respiratory complexes and supercomplexes reflect the changes at the levels of respiratory complex subunits we detected in OPA1 KO. Importantly, these data show that OPA1 KO affects neither the assemblies of individual complexes I–IV nor the formation of supercomplexes.

## OPA1 is necessary for proper assembly of respiratory complex V

Our data show that OPA1 KO has little effect on the assemblies of individual complexes I–IV. Most strikingly, however, complex V in OPA1-KO mitochondria showed a marked impairment of its assembly (Fig. 6a). Control mitochondria showed similar levels of monomer ($V_1$) and dimer ($V_2$) with additional oligomeric forms ($V_{oligo}$). In OPA1-KO mitochondria, we observed remarkable accumulations of unassembled smaller sub-complexes (Fig. 6a, b). Furthermore, there were overall decreases of monomer, dimer, and oligomer levels (Fig. 6a, b). These data suggests that OPA1 is required for proper assembly of complex V but not the other respiratory complexes. At this point, we do not know whether OPA1 is an active assembly factor of complex V or is necessary for its stability. Complex V ($F_oF_1$-ATP synthase) is composed of the catalytic $F_1$ complex that synthesizes ATP and the membrane-intrinsic $F_o$ complex containing the c-ring and the peripheral stalk with multiple membrane proteins referred to as the MPs here (Supplementary Fig. S4a). The unassembled complex V sub-complexes in OPA1-KO mitochondria were detected by the subunit c antibodies, indicating that they are sub-$F_o$ complexes containing the c subunits (Supplementary Fig. S4b). On the other hand, we detected the intact $F_1$ in OPA1-KO mitochondria using the anti-β subunit antibody (Supplementary Fig. S4c). The blots also showed an accumulation of the $F_1$ coupled to the c-ring ($F_1$/c-ring). These observations demonstrate that OPA1 is required for the $F_o$ assembly in the IM, but not for the $F_1$ assembly that occurs in the matrix. Although the $F_1$ assembly is unaffected, its accumulation in OPA1-KO mitochondria indicates that the incorporation of the $F_1$ for the formation of the holo complex V was decreased because of insufficient $F_o$ complex. These data show that the lack of the OPA1 protein impairs $F_o$ assembly, resulting in decreases of $F_o$-$F_1$ coupling and the subsequent dimer/oligomer formation.

It was indicated that OPA1 KO in skeletal muscles decreases the mitochondrial DNA (mtDNA) level, respiration, and mitochondrial mass[20,22]. While OPA1 LKO caused no functional defect of mitochondria, previous reports showed that the absence of mtDNA in rho zero cells causes assembly defects of respiratory complexes[57,58]. Therefore, we examined mtDNA in OPA1-KO liver. The quantitative PCR analyses

indicated that the level of mtDNA in OPA1 KO liver was decreased to an approximately half of the control liver (Fig. 6c), raising the possibility that the impaired complex V assembly that we observed in OPA1-KO liver might be due to the decreased mtDNA. To test whether the complex V assembly defect is a specific effect of OPA1 KO, we compared the complex V assembly in OPA1-KO and Mfn1/2-double KO (Mfn-DKO) mouse embryonic fibroblasts (MEFs). It was shown that OPA1 KO or Mfn DKO causes a decrease of mtDNA to a similar extent because of the loss of mitochondrial fusion[59]. Consistently, our data show that both OPA1-KO and Mfn-DKO MEFs have significantly decreased levels of mtDNA (Fig. 6d). We found that, despite the similar decrease in mtDNA, the complex V assembly state in Mfn-DKO cells was markedly different from that of OPA1-KO cells (Fig. 6e), showing much less unassembled c-ring while maintaining significant levels of monomeric and multimeric complex V (Fig. 6e). Mfn-DKO MEFs show a substantially lower unassembled/assembled ratio as well as higher multimer/monomer ratio, compared with those of OPA1-KO cells (Fig. 6f, g). These data suggest that the complex V assembly defect in OPA1-KO liver is likely a specific effect of OPA1 KO and, not a general phenomenon from a decrease in mtDNA. Furthermore, OPA1 depletion was shown to cause more severe respiratory defect than Mfn DKO in MEFs[60], indicating a poor correlation between mitochondrial function and mtDNA levels. Although our MEFs data suggest a specific role of OPA1 in complex V assembly, to what extent a decrease in mtDNA content in OPA1 KO liver contributes to the complex V assembly defect needs to be further investigated. While the decrease in complexes I and IV in OPA1-KO liver is suggestive of the depleted mtDNA phenotype to some degree, we observed the increase in complexes II and V in OPA1 KO liver, suggesting an involvement of stress response in the contents of the respiratory complexes. In the liver, as we observed, OPA1 KO induces an efficient ISR, which increases mitochondrial biogenesis (including mtDNA) to preserve mitochondrial function. Pronounced complex V assembly defect by OPA1 KO with a functionally inconsequential reduction in mtDNA level suggests a potentially specific role of the OPA1 molecule in the assembly or stability of complex V.

Despite the decreased level of mature complex V, OPA1-KO mitochondria exhibited an unaffected ADP-driven state 3 respiration, indicating that the reduced level of complex V is sufficient for ATP synthesis. Mitochondria can maintain respiration and ATP synthesis with a considerable reduction in respiratory complex levels, called mitochondrial threshold effect[61]. Although OPA1 KO reduces complex and supercomplex levels, our data showed that these levels of complexes are sufficient for supporting normal respiratory activity.

The previously defined ISR induced by mitochondrial stress involves the activation of the mitochondrial metalloprotease OMA1 as an upstream event leading to eIF2α phosphorylation[62,63]. Those studies used respiration inhibitors (oligomycin and CCCP) to cause respiratory defect as the mitochondrial stress. In contrast, OPA1-KO livers showed unimpaired respiration and the OMA1 inactivation (lacking activation-induced autocatalytic degradation) (Supplementary Fig. S5a)[64,65]. These observations show that the mitochondrial stress caused by OPA1 KO is not the respiratory defect. Our proteomic data showing the large increases of the Lon protease and mitochondrial chaperones along with the accumulation of unassembled complex V suggest that OPA1-KO liver induces the ISR through the mitochondrial proteostatic stress.

## OPA1-KO livers are protected from APAP-induced injury

The proteomic analyses indicate that, facing the absence of the OPA1 function in mitochondria, the liver re-establishes the mitochondrial and cellular activities through a stress response to preserve mitochondrial and liver function, which suggests the induction of mitohormesis. Mitohormesis can protect cells from future insults[28], similar to the preconditioning effect. Therefore, we tested whether OPA1 KO plays a protective role using a DILI model. Acetaminophen (APAP) overdose is the most frequent cause for liver failure in humans and is

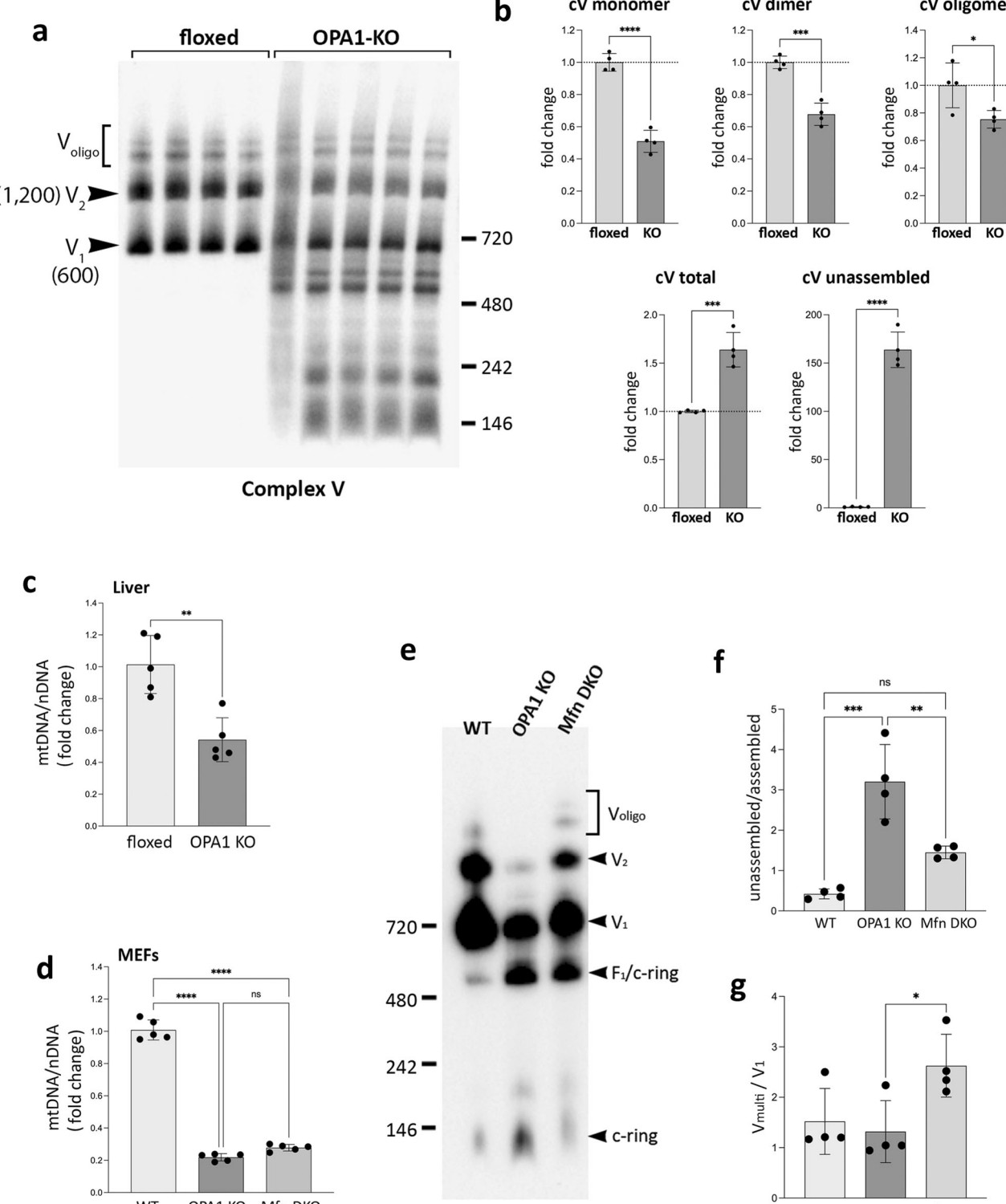

**Fig. 6 | OPA1 KO impairs complex V assembly. a** OPA1 KO significantly impairs the assembly of complex V. Sub-complexes smaller than complex V monomer ($V_1$) are accumulated in OPA1-KO mitochondria. **b** Quantification of (**a**). The increases in total complex V ($p = 0.0004$) and unassembled ($p < 0.0001$). The deceases in complex V monomer ($p < 0.0001$), dimer ($p = 0.0002$), and oligomer ($p = 0.0304$). $n = 4$ animals per group. Data are presented as mean values +/- SD. Unpaired t test. **c** A decrease of mtDNA in OPA1-KO liver. $n = 5$ animals per group. Data are presented as mean values +/- SD. Unpaired t test. **p = 0.0017. **d** Decreased mtDNA in OPA1-KO or Mfn-DKO MEFs. $n = 5$ independent measures per group. Data are presented as mean values +/- SD. Unpaired t test. ****$p < 0.0001$. **e** Complex V assembly states in WT, OPA1-KO, and Mfn-DKO MEFs by BNGE immunoblot (anti-subunit c antibodies). **f**, **g** Quantification of (**e**). $n = 4$ independent experiments per group. Data are presented as mean values +/- SD. One-way ANOVA with Turkey's multiple comparisons. ***$p = 0.0001$; **$p = 0.0036$; *$p = 0.0401$. Source data are provided as a Source Data file.

the well-established model of DILI. The APAP toxicity is mainly from conversion of APAP to the reactive species by CYP2E1, which induces oxidative stress and mitochondrial damage for hepatic injury[66–68]. Because APAP causes mitochondrial stress, we first tested whether the APAP insult itself induces the ISR. We administered APAP (350 mg/kg i.p. injection), and examined eIF2α phosphorylation. Neither eIF2α phosphorylation nor the FGF21 level was changed significantly, indicating that the ISR is not a part of liver pathophysiology in APAP overdose (Supplementary Fig. S5b).

We examined the liver injury at 6 hours post APAP administration (350 mg/kg). Remarkably, histological analyses revealed that OPA1 KO almost completely prevented APAP-induced liver injury. H&E staining showed no liver injury in OPA1-KO livers, whereas control livers had wide-spread injury displaying extensive centrilobular necrosis (Fig. 7a). Consistently, serum ALT levels were significantly low in APAP-treated OPA1-LKO mice (Fig. 7b). In APAP overdose, depletion of glutathione (GSH) and activation (phosphorylation) of JNK play important roles in the APAP injury by amplifying oxidative stress[68,69]. OPA1 KO maintained significant GSH levels and the GSH/GSSG ratio in APAP treatment (Fig. 7c). Furthermore, phospho-JNK levels and the conversion of LC3-I to LC3-II were significantly low in APAP-treated OPA1-KO livers compared with control livers (Fig. 7d and e).

APAP is converted by CYP (mostly CYP2E1) to the reactive metabolite *N-acetyl-p-benzoquinone imine* (NAPQI) that is detoxified by GSH. In APAP overdoses, NAPQI depletes GSH and the excess covalently binds to the cysteine residue of proteins to form 3-(cystein-S-yl)-acetaminophen (APAP-CYS) as adducts[70], causing mitochondrial dysfunction and cell death. Our proteomic analyses show decreases in many CYP family proteins in OPA1-KO livers (Fig. 7f), predicting a potentially beneficial effect of OPA1 KO in DILI. Importantly, we found a significant decrease of CYP2E1 in OPA1 KO livers (Fig. 7g), which would decrease the conversion of APAP to NAPQI. However, our data show that APAP overdose to OPA1-LKO mice still decreased GSH levels (Fig. 7c), suggesting significant generation of NAPQI and possibly the APAP-CYS adduct. Therefore, we analyzed the levels of APAP-CYS in liver. APAP-CYS is shown to be stable in mouse liver for at least 6 hours after APAP treatment[71]. In saline-treated mice, no APAP-CYS was detected, as expected (Fig. 7h, ND). APAP-treated floxed mice showed a large increase of APAP-CYS, causal for liver injury (Fig. 7h). We found that OPA1 KO significantly decreased APAP-CYS in the liver (Fig. 7h), supporting the protective effect of OPA1 LKO. However, the decreased level of APAP-CYS in OPA1-KO liver was still considerable, which was found to be similar to that formed with the 150 mg/kg APAP treatment that caused a moderate injury[71]. Because our data indicate no liver injury with a substantial formation of the APAP-CYS adduct in OPA1 KO liver, the decreased CYP2E1 may not be the sole mechanism of the protection by OPA1 LKO.

## OPA1-KO mitochondria maintain mitochondrial function under APAP toxicity

In APAP overdose, the liver injury occurs mainly through mitochondrial ROS overproduction caused by APAP-protein adduct and mitochondrial permeability transition (MPT), leading to necrosis of hepatocytes[67,72]. Therefore, we examined the mitochondrial function in APAP treatment. To test the effect of OPA1 KO on APAP-induced mitochondrial dysfunction, we analyzed mitochondrial membrane potential (MMP), respiration, and the MPT sensitivity of liver mitochondria from APAP-treated control and OPA1-KO mice. In the MMP evaluation using the rhodamine 123 quenching assay, liver mitochondria from APAP-injected control floxed mice were unable to maintain the MMP, demonstrating that the APAP toxicity impaired the mitochondrial electron transport activity. In contrast, OPA1-KO mitochondria maintained the normal level of MMP (Fig. 8a, b), demonstrating preservation of mitochondrial function by OPA1-KO under APAP insult. OCR analyses also showed that APAP insult has no

effect on respiration activity in OPA1-KO mitochondria, whereas it impairs respiration in control mitochondria (Fig. 8c, d). Mitochondrial Ca²⁺ overload is the main effector for MPT. Therefore, the MPT sensitivity can be assessed by mitochondrial Ca²⁺ retention capacity (mCRC) assay, in which repeated Ca²⁺ pulses are added until mitochondria no longer take up Ca²⁺[73]. We found that APAP decreased mCRC in floxed control mice, demonstrating that liver mitochondria from APAP-treated mice are predisposed to Ca²⁺-induced MPT. However, OPA1-KO mitochondria from APAP mice maintained the mCRC, nearly identical to those from saline-injected mice with and without the MPT inhibitor cyclosporine A (CsA) (Fig. 8e, f). This series of data demonstrates that OPA1 KO protects liver mitochondria from APAP-induced mitochondrial injury. Furthermore, these data show that, in addition to the decreased CYP2E1, OPA1 KO provides the protection at the mitochondrial phase of APAP-induced liver injury by preserving mitochondrial function.

## OPA1 KO confers resistance to MPT and lowers mitochondrial ROS production

Our results suggest that OPA1-KO mitochondria are less sensitive to MPT. Therefore, we tested Ca²⁺-induced mitochondrial swelling with and without CsA. Alamethicin was added at the end to induce the maximum swelling. In both conditions, we found that OPA1-KO mitochondria exhibited not only slower swelling, but also markedly decreased swelling compared with control mitochondria (Fig. 9a-c). These data demonstrate that OPA1 KO confers resistance to Ca²⁺-induced MPT. We further examined characteristics of mitochondrial Ca²⁺ uptake to explore the potential mechanisms. We found that OPA1 KO mitochondria take up Ca²⁺ faster than control mitochondria (Fig. 9d). Adding higher concentration of Ca²⁺ induced the MPT, releasing Ca²⁺ after the initial Ca²⁺ uptake (Fig. 9e). Under this condition, OPA1-KO mitochondria exhibited a delayed onset of MPT with a lesser extent. ROS is a main permissive factor for MPT, and we examined mitochondrial ROS production. We found that the rate of ROS production in OPA1-KO mitochondria was 5-fold lower than that of control mitochondria (Fig. 9f–h).

Immunoblotting revealed a marked increase in the level of the mitochondrial Ca²⁺ influx channel MCU in OPA1 KO with a minor increase of the efflux channel NCLX (Fig. 9i, j), potentially supporting the increased Ca²⁺ uptake. Interestingly, OPA1 KO greatly increased the CypD level, which supposedly increases the MPT sensitivity as opposed to what we observed. This result suggests that the decreased MPT sensitivity in OPA1 KO is independent of the MPT pore. CypD is the peptidylprolyl isomerase F, a mitochondrial chaperone; therefore, its increase is likely from OPA1 KO-induced proteostatic stress response. The mitochondrial antioxidant MnSOD was significantly increased, potentially explaining the decrease in mitochondrial ROS production in OPA1 KO (Fig. 9i, j). There was no significant change in GPx1/2 levels. These data revealed that OPA1 KO-induced ISR brings about an intrinsic mitochondrial property change, conferring resistance to MPT and generating less ROS. Mechanistically, our results demonstrate that OPA1 KO protects the liver at both initial (by decreasing the CYP2E1 level) and subsequent (by mitochondrial reinforcement) stages of APAP toxicity.

In addition to DILI, we also tested the effect of OPA1 LKO on metabolic burden by diet-induced obesity model. Body weight of the floxed control mice during the 12-week high fat diet (HFD) increased significantly (supp. Fig. S6a). Remarkably, however, OPA1-LKO mice showed no gain in body weight in HFD. Accordingly, OPA1 LKO mice showed significantly faster blood glucose clearance in GTT (supp. Fig. S6b, c). Consistently, H&E and Oil Red O staining revealed greatly decreased steatosis in OPA1-KO liver in HFD whereas marked steatosis in floxed liver was observed (supp Fig. S6d). These results indicate that OPA1-LKO mice are resistant to HFD-induced hepatic steatosis and obesity.

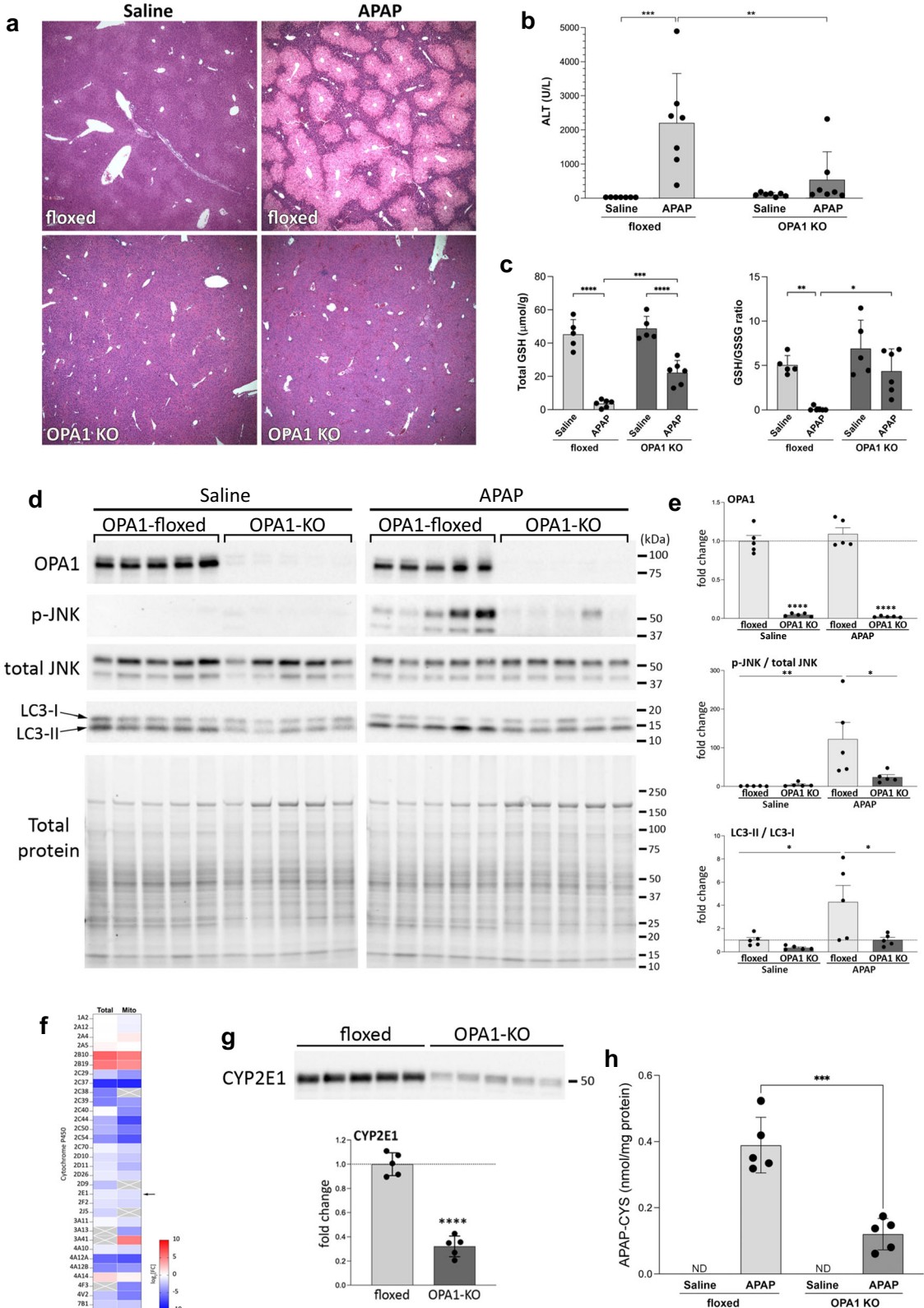

**Fig. 7 | OPA1-KO livers are protected from APAP-induced injury. a** H&E staining of liver sections from control and OPA1-KO mice with and without APAP. Representative images from at least 7 animals per group. The results were consistent. **b**, **c** OPA1 KO mitigates the ALT increase and GSH depletion in APAP treatment. *n* = 7 (**b**, ALT), 5 (saline in **c**) and 6 (APAP in **c**) animals. Data are presented as mean values +/- SD. Two-way ANOVA with Turkey's multiple comparisons. ***p = 0.0003; **p = 0.0054 (**b**). ****p < 0.0001; ***p = 0.0007; **p = 0.0046; *p = 0.0102 (**c**). **d**, **e** Immunoblots for p-JNK and LC3 in floxed and OPA1-KO livers with and without

APAP. *n* = 5 animals per group. Data are presented as mean values +/- SD. Two-way ANOVA with Turkey's multiple comparisons. ****p < 0.0001 (OPA1); **p = 0.0066 (p-JNK); *p = 0.0291 (p-JNK), 0.0281 (LC3), and 0.0284 (LC3). **f** A heatmap of CYPs found in total and mitochondrial proteomes. Arrow: CYP2E1. **g** CYP2E1 immunoblot. Total protein is shown in Fig. 9i. *n* = 5 animals per group. Data are presented as mean values +/- SD. Unpaired t test. *p* < 0.0001. **h** Quantification of liver APAP-CYS; *n* = 5 animals per group. Data are presented as mean values +/- SD. Unpaired t test. *p* = 0.0003. Source data are provided as a Source Data file.

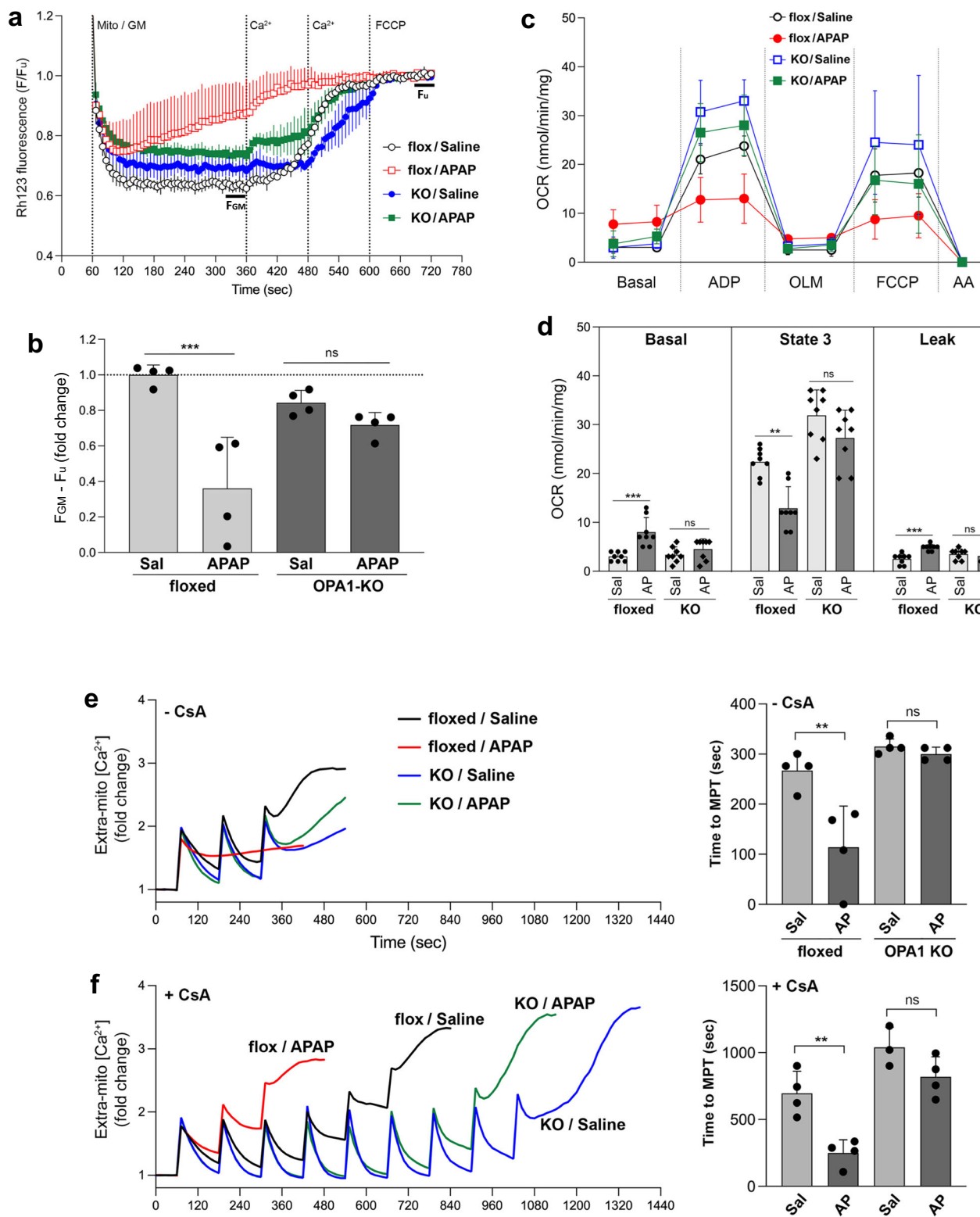

**Fig. 8 | OPA1-KO mitochondria maintain mitochondrial function under APAP toxicity. a, b** MMP assessment by Rhodamine 123 quenching assays. Representative traces (**a**) and quantification (**b**). $n = 4$ animals per group. Data are presented as mean values +/- SD. Two-way ANOVA with Turkey's multiple comparisons. ***$p = 0.0004$. **c, d** OCR measurements (**c**) and quantification (**d**). $n = 4$ animals per group over 2 measures. Data are presented as mean values +/- SD. Two-way ANOVA

with Turkey's multiple comparisons. ***$p = 0.0002$ (basal); **$p = 0.0019$ (state 3); ***$p < 0.0003$ (leak). **e, f** mCRC assays. Representative traces and quantification without (**e**) and with CsA (**f**). $n = 4$ animals per group. Data are presented as mean values +/- SD. Two-way ANOVA with Turkey's multiple comparisons. **$p < 0.0025$ (**e**); **$p = 0.0048$ (**f**). Source data are provided as a Source Data file.

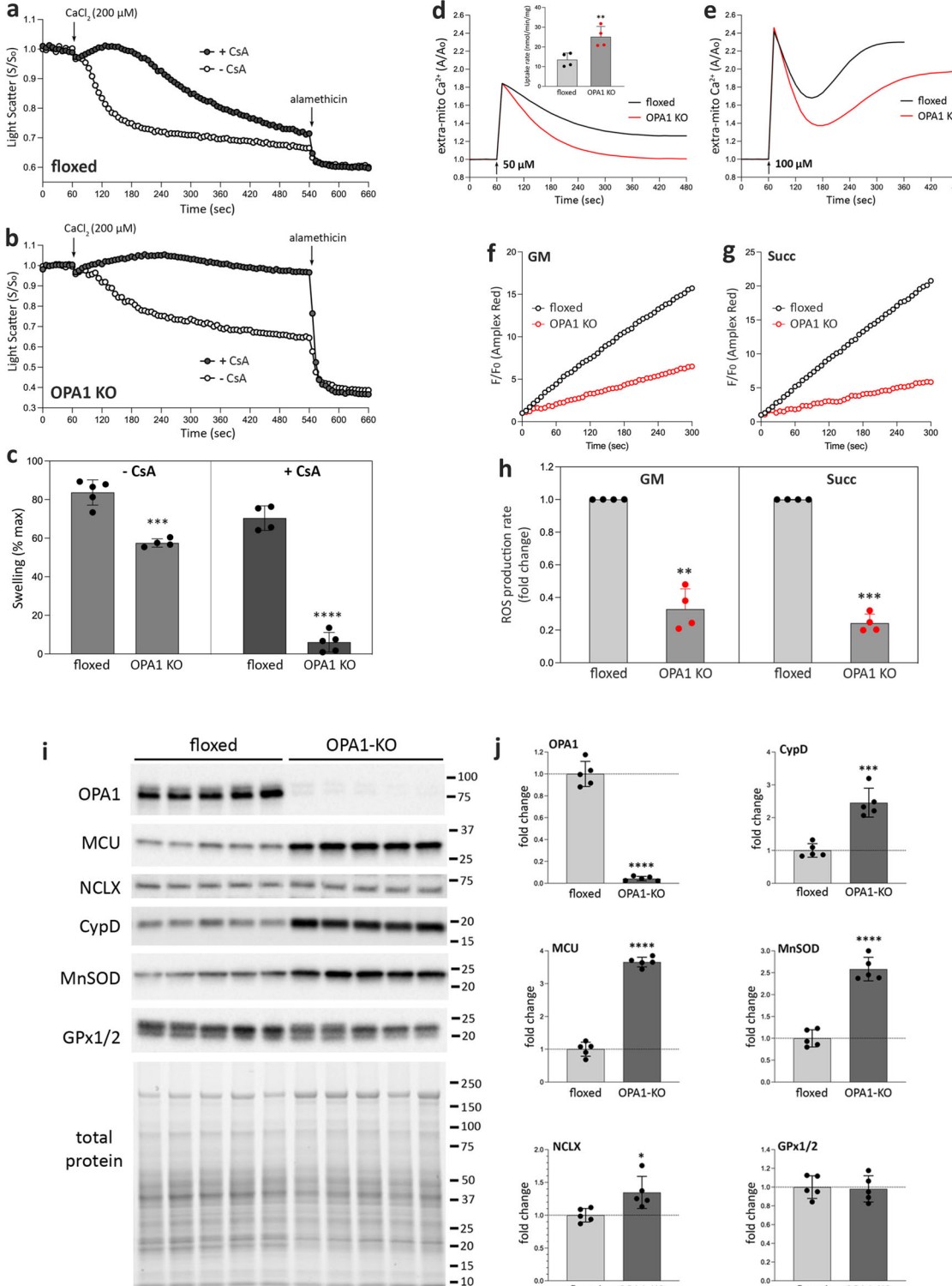

**Fig. 9 | OPA1 KO increases the resistance to MPT and lowers mitochondrial ROS production. a–c** Mitochondrial swelling by light scattering assays. Swelling profiles of control (**a**) and OPA1-KO (**b**) mitochondria with and without CsA, and quantification (**c**). $n = 4$ animals per group. Data are presented as mean values +/- SD. Unpaired t test. ***$p = 0.0001$; ****$p < 0.0001$. **d**, **e** Mitochondrial Ca²⁺ uptake profiles with 50 μM (**d**) and 100 μM (**e**) CaCl₂ added to the mitochondrial suspension. The inset in (**d**) is the calculated Ca²⁺ uptake rate (nmol/min/mg mito). $n = 4$ animals

per group. Data are presented as mean values +/- SD. Paired t test. **$p = 0.0036$. **f–h** Mitochondrial ROS production with glutamate/malate (**f**, GM) and succinate (**g**) as substrate, and quantification (**h**). $n = 4$ animals per group. Data are presented as mean values +/- SD. Paired t test. **$p = 0.0017$; ***$p = 0.0001$. **i** OPA1 KO increases the levels of MCU, NCLX, CypD, and MnSOD. **j** Quantification of (**i**). $n = 5$ animals per group. Data are presented as mean values +/- SD. Unpaired t test. ****$p < 0.0001$; *$p = 0.0189$; ***$p = 0.0002$. Source data are provided as a Source Data file.

These protective effects of OPA1 KO on DILI and metabolic burden indicate the mitohormetic effect brought about by ISR. The liver resiliency is mainly attributed to liver's high regenerative capacity after injury. As our data show that OPA1 KO induces mitohormesis to protect liver from drug toxicity and steatosis, the liver mitohormesis can be considered another mechanism contributing to the liver resiliency.

## Discussion

OPA1 is important for maintaining mitochondrial function. As its name indicates, optic atrophy is the hereditary disorder that has led to the identification of the OPA1 gene in humans[74,75]. Human OPA1 mutations cause autosomal dominant optic atrophy (ADOA) and ADOA plus, depending on the severity of OPA1 defects[74–76]. The clinical spectrum of OPA1 mutations has been expanding beyond the visual impairment. ADOA plus patients exhibit vision and hearing losses that are often accompanied with myopathy, ataxia, and peripheral neuropathy, cardiomyopathy, and developmental delay[76–80]. The liver is central to metabolic regulation that requires proper mitochondrial activity. However, hepatic phenotypes have not been reported with ADOA plus, except for one rare case in compound heterozygous OPA1 mutations with two additional homoplasmic mitochondrial DNA mutations[80]. Our study showed that OPA1 is dispensable in the liver, potentially explaining the scarcity of the hepatic phenotype in ADOA plus.

We observed disrupted cristae structure in OPA1-KO liver mitochondria as expected. In some mitochondria, cristae lost their cristae junction, while in others, there were many short primordial cristae that have the cristae junctions. However, junction-less cristae still maintained their narrowness in OPA1-KO mitochondria, suggesting that OPA1 is not the structural requirement for cristae tightness. Furthermore, the EM of isolated mitochondria from normal liver showed widened cristae that are much wider than those formed by the suggested OPA1 cross-bridge. Interestingly, these cristae often had sharp angled ridges that were not found in OPA1-KO mitochondria, suggesting that OPA1 may provide a structural constraint in the cristae membrane rather than forming intracristal cross-bridges. We found that OPA1 KO causes a defect in complex V formation, resulting in the decreases of dimeric and oligomeric complex V that shape cristae ridge. It is possible that OPA1 is involved in cristae shaping rather indirectly by regulating complex V levels. Alternatively, prohibitins may participate in regulating cristae shape. Prohibitins are known to form ring complexes in the inner membrane, functioning as scaffolds for lipids and proteins, which can provide shape constraint in the cristae[81]. Prohibitins are suggested to control the OPA1 function by membrane organization of m-AAA proteases[82]. Prohibitin may interact with OPA1[18]. It is possible that OPA1 KO may affect the prohibitin structure to disturb cristae morphology.

OPA1 is known as a dual function protein that mediates IM fusion and cristae maintenance. Our data showed that OPA1 has an additional role in complex V assembly. Although we and others previously observed an increased presence of free $F_1$ in OPA1 deletion in fibroblasts[17,18], the current study demonstrated that OPA1 is important for $F_o$ assembly specifically. Our data indicate that OPA1 KO causes a reduced level, not a complete loss, of mature complex V. As such, it is possible that OPA1 indirectly affects complex V assembly through its membrane remodeling activity or providing the stability of the holo complex V. This effect of OPA1 KO supports the notion that the liver ISR induced by OPA1 KO is different from that by respiratory deficiency. We also found inactivation of OMA1 in OPA1-KO liver. The main substrate of OMA1 is OPA1[83,84]. It is possible that, in OPA1 KO, OMA1 is inactivated/stabilized without the autocatalytic degradation because of the absence of its main substrate. The accumulation of unassembled complex V sub-complexes in OPA1 KO suggests that the liver ISR in OPA1 KO is induced by mitochondrial proteostatic stress, and does not involve the OMA1-mediated process.

The observed halt in age-associated weight gain in OPA1-LKO mice shows that the ISR changes whole body metabolism. These global changes would ultimately allow an establishment of a new homeostatic state for sustained functioning of the liver. Our results indicate that the OPA1 KO-induced stress response extends its capacity beyond the basic preservation of mitochondrial function to the mitohormetic protective response. The mitohormetic response by OPA1 KO is likely liver specific, as OPA1 KOs in other tissues are harmful or lethal. OPA1 depletion in neurons and fibroblasts was shown to cause mitochondrial deficiency, decreasing the mCRC and increasing ROS production and the sensitivity to oxidant insult[35,85]. However, OPA1 KO in the liver rather strengthens mitochondria, despite the complex V assembly defect, indicating that mitochondrial proteostatic stress is causal for the mitochondrial reinforcement for mitohormetic effect.

The dispensable nature of OPA1 and the protective effect of its KO are based on the study with 8-12 week KO mice, and it is unclear whether extended OPA1 KO maintains the same effect. We found that the halt in weight gain after AAV-Cre injection is a reliable indicator of the efficient KO of OPA1 and ISR induction. A long-term monitoring of mice injected with AAV8-TBG-Cre showed that body weight started to increase after 16 weeks, indicating that AAV-induced Cre expression becomes ineffective after 16 weeks (Supplementary Fig. S7a). Indeed, there was a significant recovery of OPA1 expression in the liver of AAV-Cre-injected mice after 24 weeks (Supplementary Fig. S7b). While this limits the study for the long-term effect of OPA1 LKO, 8–12 weeks KO is likely sufficient to test the OPA1-KO effect. For example, acute muscle OPA1-KO mice die in 8–12 weeks[22].

The protection from APAP toxicity by OPA1 KO is remarkable, showing near complete prevention of injury. Our data indicate that the protection mechanisms in OPA1 KO include the intrinsic change of mitochondrial property and the reduced level of CYP2E1. Considering significant level changes in the proteins involved in these processes, the protection by OPA1 KO is likely the consequence from the stress response. We found that the OPA1 KO by Alb-Cre incompletely deletes the OPA1 and shows no protection from APAP toxicity and minimal ISR induction (Supplementary Fig. S7c-f), suggesting that incomplete KO or a developmental adaptation to the reduced OPA1 level may prevent the robust ISR induction. These observations indicate that the ISR is a critical factor for the mitohormetic response in OPA1 KO. While our data point to the ISR being the major factor contributing to the mitohormetic effect of OPA1 LKO, more direct evidence would be necessary to confirm this notion. FGF21 is a major ISR effector, and FGF21 KO was previously shown to abolish the metabolic benefit induced by liver ISR in HFD[86]. It will be interesting to test if FGF21 KO eliminates the protective effect of OPA1 LKO in APAP toxicity. To directly test the role of ISR, the chemical inhibitor of ISR, ISRIB can also be considered[87].

Previous studies showed that OPA1 depletion attenuates liver damage in mouse model of NASH[27]. Those studies indicated that OPA1 depletion blocks mitochondrial fusion and thus prevents the formation of mega-mitochondria in NASH, which in turn increases mitophagy to mitigate the liver pathology. However, our studies demonstrated that OPA1 KO induces large-scale changes in liver and mitochondrial activities through a mitohormetic response for protective effect. The liver has high regenerative capacity after injury, which is the major mechanism of liver resiliency. As the mitohormetic response induced by OPA1 KO not only preserves liver function but also protects the liver from a second insult, the mitohormesis is likely a previously unrecognized mechanism for the liver resiliency.

## Methods

### Study approval

All animal experiments were performed according to procedures approved by the IACUC at Augusta University.

## Generation of liver-specific OPA1 knockout mice and animal experiments

Mice were kept under controlled temperature and lighting (20–22 °C; 12-h dark-light period) in 50% humidity and with free access to a standard chow diet (Teklad global rodent diet, 2918) and water.

OPA1-LKO mice were generated by administering AAV8–thyroid-binding globulin (TBG)-Cre to OPA1[flox/flox] mice. Generation of OPA1[flox/flox] mice were described previously[54]. OPA1[flox/flox] mice were generated in a C57BL/6-129/SvEv mixed background. AAV8-TBG-GFP was used to generate OPA1-floxed control mice. Both AAV8-TBG-Cre (AAV.TBG.-PI.Cre.rBG) and AAV8-TBG-GFP (AAV.TBG.PI.eGFP.WPRE.bGH) were obtained from Addgene. AAV.TBG.PI.Cre.rBG and AAV.TBG.-PI.eGFP.WPRE.bGH were gifts from James M. Wilson (Addgene viral prep number # 107787-AAV8 and # 105535-AAV8, respectively). Eight-week-old OPA1[flox/flox] male mice were given an intraperitoneal (i.p.) injection of $1.5 \times 10^{11}$ genome copies of AAV8–TBG-Cre or AAV8–TBG-GFP. The i.p. injection of AAV has been shown to be equally effective to the intravenous or intraportal injection for liver-specific expression[88,89]. Male mice were used because female mice show inconsistency in AAV-mediated expression in the liver[88].

Body weights were recorded on a weekly base at the same time of the day (between 4 and 5 PM). Mice were analyzed 8–12 weeks after the AAV administration. For glucose tolerance tests (GTT), overnight-fasted mice received i.p. injections of 2 mg glucose/g body weight. Tail blood glucose levels were measured using a blood glucose meter before and at 15, 30, 60 and 120 min after glucose injection. For metabolic parameter measurements, each mouse was singly housed in the Comprehensive Lab Animal Monitoring System (CLAMS, Columbus Instruments). Seventy two-hour measurements were processed for average daily and hourly values of $VO_2$, $VCO_2$, RER, EE, and food consumption. The ALT and AST activity levels in mouse serum were measured according to the manufacturer's instructions (Cayman Chem.). The serum LDH activity level was measured according to the manufacturer's instructions (Biorbyt). For histology, liver tissues were fixed in 4% paraformaldehyde in phosphate buffered saline, paraffin embedded, sectioned and stained with Hematoxyline-Eosin (H&E). For HFD, the diet containing 60% of daily caloric intake by fat (0.82 kcal/g protein; 3.24 kcal/g fat; 1.43 kcal/g carbohydrate) (Bioserv S3282) was supplied ad libitum to floxed and OPA1-LKO mice at 1 week post AAV-Cre administration for 12 weeks. For Oil red O staining, liver tissues were fixed in 4% paraformaldehyde in phosphate buffered saline, then changed to 30% sucrose overnight, frozen sectioned and stained with Oil Red O.

## Isolation of mitochondria

Mitochondria were isolated from the mouse livers by differential centrifugation. Mouse liver was homogenized using a Potter-Elvehjem homogenizer (5 strokes) in an ice-cold sucrose buffer containing: 200 mM sucrose, 10 mM Tris–MOPS, pH7.4 and 0.1 mM EGTA-Tris. The homogenate was then centrifuged at 800 x g for 10 min to remove debris. The supernatant was centrifuged at 8,000 x g for 10 min to precipitate mitochondria. The pellet was washed twice by centrifugation at 8,000 x g for 10 min using sucrose buffer. A final mitochondria-enriched pellet was re-suspended in sucrose buffer. The amount of mitochondrial protein was determined using the Bradford protein assay (Bio-Rad). Isolated mitochondrial samples were kept on ice for respiration analyses. Aliquots of the isolated mitochondria were stored in −80 °C.

## Transmission electron microscopy

The livers of anesthetized mice were flushed with 0.1 M sodium cacodylate and perfused with 4% PFA and 2.5% glutaraldehyde in 0.1 M sodium cacodylate pH 7.4. Matching lobes of the liver of control and OPA1-LKO mice were cut down to 3-4 mm tissue slabs and were incubated in fixative for 24 hours. Corresponding sections of the control and KO livers were further dissected into approximately 1 mm cubes. The hepatic cubes were then post-fixed in 1% osmium tetroxide and 1.5% potassium ferrocyanide in 0.1 M sodium cacodylate for 120 minutes. Following 2 ten-minute washes in ddH₂O, the tissue was dehydrated stepwise from 50% ethanol to 100%. Samples were then transition to propylene oxide, and infiltrated with Epon 812/araldite resin (Electron Microscopy Sciences, EMS) overnight. Samples were placed in fresh resin and embedded by incubating at 65 °C for 48 hours. Blocks were thin-sectioned at ~55 nm with a Leica UC-7 ultramicrotome (Leica) and mounted to formvar carbon coated slot grids (EMS). Prepared grids were stained with 2% uranyl acetate and 0.3% lead citrate prior to imaging on a Hitachi 7650 transmission electron microscope (Hitachi) equipped with an Erlangshen 11 MP camera (Gatan) using DigitalMicrograph software (Gatan).

For EM of isolated mitochondria, isolated mitochondria were fixed by resuspending in room temperature fixative, 4% PFA and 2.5% glutaraldehyde in 0.1 M sodium cacodylate pH 7.4. Fixed mitochondria were washed twice for 10 minutes in 0.1 M sodium cacodylate and were pelleted at 1600 x g for 3 minutes. Pelleted mitochondria were then post-fixed by resuspension in 1% osmium tetroxide and 1.5% potassium ferrocyanide in 0.1 M sodium cacodylate for 40 minutes, washed twice in ddH₂O and then trapped in 3% agarose. The agarose was allowed to solidify at 4 °C. Approximately 1 mm agarose cubes were then processed for embedment into Epon 812/araldite resin. Samples were dehydrated, infiltrated and embedded as above. Blocks were sectioned and imaged as above. Processing for and image acquisition using transmission electron microscopy was conducted at the University of Rochester Electron Microscopy Resource in the Center for Advanced Research Technologies.

## Respiration analyses

Oxygen consumption rates (OCRs) of isolated mitochondria were measured using an XFe24 Analyzer (Agilent). Isolated mitochondria (10-20 µg) were transferred to an XFe24 microplate in mitochondrial assay solution (70 mM sucrose, 220 mM mannitol, 10 mM KH₂PO₄, 5 mM MgCl₂, 2 mM HEPES, 1 mM EGTA, 0.2% (w/v) fatty acid-free BSA, pH 7.2) containing 5 mM glutamate / 2.5 mM malate. Plates were spun at 2000 x g for 20 minutes at 4 °C for attachment of mitochondria. For OCR, 4 mM ADP, 3 µM oligomycin, 4 µM carbonyl cyanide p-trifluoromethoxyphenylhydrazone (FCCP), and 4 µM antimycin A were injected in sequence. OCR was normalized by protein amount.

## ATP assay

ATP was extracted from liver tissues (30 mg) by homogenizing in 100 µl ice-cold Tris EDTA-saturated phenol using 0.5 mm glass beads[90]. Homogenate was transferred into microtubes containing 20 µl chloroform and 100 µl water. The homogenate was shaken thoroughly for 20 s and centrifuged at 10,000 x g for 5 min at 4 °C. The supernatant was diluted with water and ATP content was measured using ATP Detection Assay kit (Cayman) on a luminescence microplate reader.

## Immunoblotting

Protein samples were prepared from liver homogenates in Laemmli sample buffer, run on SDS-polyacrylamide gels (4-15% TGX stain-free gel, Bio-Rad), and transferred to the polyvinylidene difluoride (PVDF) membrane. The membranes were blocked, incubated with primary antibodies overnight at 4 °C, followed by secondary antibodies, and developed using the chemiluminescence imaging system (Bio-Rad). Following primary antibodies were used: OPA1 (BD Biosciences, 612606; 1:1000); caspase-3 (Cell Signaling, 9662; 1:1000), PARP-1 (Cell Signaling, 9542; 1:1000), β-actin (Sigma, A1978; 1:40000), TOM20 (Proteintech, 11802-1-AP; 1:1000), cytochrome c (BD Biosciences, 556432; 1:5000), elF2α (Cell Signaling, 9722; 1:1000), phospho-elF2α (Cell Signaling, 9721; 1:1000), FGF21 (Proteintech, 26272-1-AP; 1:1000),

LC3 A/B (Cell Signaling, 4108; 1:1000), PGC1α (Invitrogen, PA5-38022; 1:500), OMA1 (Santa Cruz, sc-515788; 1:100), and mitochondria total OXPHOS rodent WB cocktail (Abcam, ab110413; 1:1000). JNK (Cell Signaling, 9252; 1:1000), p-JNK-Thr183/Tyr185 (Cell Signaling, 9255; 1:500), MCU (Sigma, HPA016480; 1:1000), NCLX (Proteintech, 21430-1-AP; 1:1000), CypD (Proteintech, 18466-1-AP; 1:1000), MnSOD (BD Biosciences, 611580; 1:1000), GPx1/2 (Santa Cruz Biotechnology, sc-133160; 1:200), and CYP2E1 (Proteintech, 19937-1-AP; 1:300).

## Proteomics and analyses

Protein extracts (50 μg) were reduced with dithiothreitol, alkylated using iodoacetamide, and digested overnight using trypsin (Thermo Scientific #90057). Digested peptides were cleaned using C18 spin column (Harvard Apparatus #744101) and then lyophilized. Peptide digests were analyzed by liquid chromatography-tandem mass spectrometry (LC-MS/MS) on an Orbitrap Fusion tribrid mass spectrometer (Thermo Scientific) coupled with an Ultimate 3000 nano-UPLC system (Thermo Scientific). Reconstituted peptides were first trapped and washed on a Pepmap100 C18 trap (5 μm, 0.3 × 5mm), and then separated on a Pepman 100 RSLC C18 column (2.0 μm, 75-μm × 150-mm) using a gradient of 2 to 40% acetonitrile with 0.1% formic acid over 40 min at a flow rate of 300 nl/min and a column temperature of 40 °C. Samples were analyzed by data-dependent acquisition in positive mode using Orbitrap MS analyzer for precursor scan at 120,000 FWHM from 300 to 1500 m/z and ion-trap MS analyzer for MS/MS scans at top speed mode (3-second cycle time). Higher-energy collisional dissociation was used as fragmentation method. Normalized collision energy was set to 35%. Dynamic exclusion was set to exclude after 1 time occurrence for 15 seconds. Raw MS data were processed via the Proteome Discoverer software (ver 1.4) and submitted for SequestHT search against the SwissProt mouse database. The percolator peptide spectrum matching (PSM) validator algorithm was used for PSM validation. Proteins unable to be identified distinguished based on LC-MS/MS analysis. The database search results alone were grouped to satisfy the principles of parsimony. A protein report was generated containing the identities and number of PSM for each protein group, which were further utilized for spectral counting based semi-quantitative analysis.

Difference in PSM between control and OPA1-KO groups was tested using negative binomial regression model for both total extract and mitochondrial fraction samples utilizing DESeq2 package in R[91]. The Benjamini-Hochberg corrected p-value with significance level of 0.05 and fold change of 2 was used to detect the differentially expressed proteins. Enrichment analysis on the significant proteins was performed in order to obtain molecular function and biological process using limma package in R[92]. Further, pathway analysis was performed on the differentially expressed proteins using QIAGEN Ingenuity Pathway Analysis[93].

## Blue native gel electrophoresis and immunoblotting

Aliquots of isolated mitochondria were resuspended in the sample buffer (50 mM BisTris, 50 mM NaCl, 10% w/v glycerol, 0.001% Ponceau S, pH 7.2) with 0.5% digitonin at 2.5 μg protein/μl (a ratio of 2 g digitonin/g of protein) for complex V detection and 1.5% digitonin (a ratio of 6 g digitonin/g of protein) for other respiratory complexes, and incubated for 15 min on ice. The mixture was spun at 20,000 x g for 30 min. ServaBlue G 250 was added to the supernatant to one fourth of the digitonin concentration and 20 μg of protein was loaded on a blue native gel (NativePAGE 3–12% Bis-Tris protein gel, Invitrogen). Gel running and transfer of the native gels to PVDF membrane were done according to the manufacture's instruction (Invitrogen). After proteins were blotted, the membranes were fixed in 8% acetic acid for 15 min. Blots were probed with following specific antibodies: NDUFA5 (GeneTex, GTX111016; 1:500) for complex I, succinate dehydrogenase subunit B (GeneTex, GTX113833; 1:500) for complex II, UQCRC2 (GeneTex,

GTX114873; 1:1000) for complex III, COX4 (GeneTex, GTX114330; 1:1000) for complex IV, and ATP5G1/G2/G3 (c subunit, Abcam, ab180149; 1:1000) and ATP synthase subunit β (Molecular Probes, A-21351; 1:500) for complex V.

## SCAF1 analysis

For SCAF1 isoform analysis, the SCAF1 genomic region was amplified using the following primers: for 113-AA SCAF1 (long isoform), the common forward primer 5′-AAGAGGGAGTCAGATCTTGTTACG-3′ and reverse primer 5′-AAGGCCTCGTTTCAGGTGGATGGG-3′ (262-bp PCR product), and for 111aa SCAF1 (short isoform), the common forward primer and reverse primer 5′-AAGGCCTCGTTTCAGGTGGAAACC-3′ (256-bp PCR product). Amplification products were analyzed by electrophoresis on agarose gels.

## mtDNA / nDNA assay

Total DNA from liver and MEFs was extracted using Extracta DNA Prep for PCR (Quantabio) as described by the manufacturer. mtDNA/nDNA ratios were measured by real-time quantitative PCR (AriaMx Real-Time PCR System, Agilent) using the SYBR-Green assay with primers for ND6 for mtDNA and GAPDH for nDNA. mtDNA/nDNA was calculated using the $2^{-\Delta\Delta Ct}$ method. ND6 primers: 5′-CCCAGCTACTACCATCATTCA AGT-3′ and 5′-GATGGTTTGGGAGATTGGTTGATGT-3′. GAPDH primers: 5′-GGCTCCCTAGGCCCCTCCTG-3′ and 5′-TCCCAACTCGGCCC CCAACA-3′[94].

## MEF culture and sample preparation

OPA1- KO MEF line was from American Type Culture Collection (ATCC CRL2995; originally generated by David Chan, Caltech). WT and Mfn-DKO MEF lines were kind gifts from David Chan (Caltech)[60]. MEFs were maintained in complete media (DMEM high-glucose medium with 10% fetal bovine serum, 1x nonessential amino acids, 100 units/ml penicillin, and 100 μg/ml streptomycin) at 37 °C in a humidified atmosphere containing 5% CO2. For blue-native gel, MEFs in culture were washed twice in ice-cold PBS, collected using a cell lifter, and pelleted at 100 × g for 3 min. Pellets were frozen at −80 °C. Pellets were resuspended in the sample buffer with 0.5% digitonin at 2.5 μg protein/μl (a ratio of 2 g digitonin/g of protein) for complex V detection.

## Acetaminophen (APAP) treatment

Mice were fasted overnight and were injected intraperitoneally with 350 mg/kg APAP diluted in warm saline or saline alone as a control. At 6 hours post APAP administration (3 hours for the mitochondrial function studies), blood was collected by cardiac puncture, and serum was prepared. The liver was quickly excised and weighed. Liver sections were fixed in 4% phosphate-buffered paraformaldehyde for histological analyses. The remaining liver was processed for mitochondrial isolation for respiration and other assays. A part of the liver was snap-frozen in liquid nitrogen and stored at −80 °C for further analyses.

## Glutathione assays

Concentrations of total GSH and disulfide dimer GSSG were measured using a glutathione assay kit (Cayman Chemical #703002). Liver tissues were homogenized in ice-cold homogenization buffer containing 50 mM phosphate pH 6.9 and 1 mM EDTA. Samples were centrifuged at 10,000 × g for 15 min at 4 °C. Collected supernatants were subjected to deproteinization by adding an equal volume of 10% metaphosphoric acid. Supernatants were collected after centrifugation and analyzed for total GSH and GSSG as per manufacturer's instructions. Total GSH in the samples was normalized with protein. Protein estimation was carried out using a BCA protein assay kit (Pierce). Free glutathione (GSH) concentration was calculated by subtraction of GSSG (1 GSSG = 2 GSH) from total GSH.

## APAP-CYS assay

The APAP-CYS content in liver was measured by LC-MS/MS following the previously described method[71]. Liver protein extract was filtered through a desalting column (Zeba spin column, Thermo Fisher) pre-equilibrated with 50 mM $(NH_4)_2HCO_3$ following the manufacturer's instructions. An aliquot (45 µl) of the filtrate was mixed with 5 µl of protease type XIV (80 U/ml) (Sigma) and incubated for 24 hours at 37 °C to liberate APAP-CYS. After digestion, 5 µL of norbuprenorphine-d3 (100 µg/ml, Cerilliant Corp.) was added as an internal standard, followed by acetonitrile (300 µL) for precipitation. Dry pellet was obtained by evaporation at 13 psi, and was reconstituted in 100 µL of 2% acetonitrile buffer with 0.1% formic acid. The reconstituted solution was cleared by centrifugation and the supernatant were transferred to a clean vial for LC-MS/MS (Thermo TSQ Quantiva triple-quadrupole mass spectrometer coupled with a Shimadzu Nexera UHPLC). Free APAP-CYS (m.w. 270, Cayman #26388) was used for the quantification standard.

## MMP evaluation

Rhodamine123 quenching assay was used for MMP evaluation. Fluorescence of rhodamine123 (2 µM; ex/em: 503/527 nm) was collected (APD130A2 Avalanche Photodetector, Thorlabs Inc.) in the respiration buffer (RB: 125 mM KCl, 2 mM $K_2HPO4$, 1 mM $MgCl_2$, 20 mM HEPES, pH 7.0) plus respiration substrate (5 mM glutamate / 2.5 mM malate) upon an addition of mitochondria (250 µg) for 5 min before the 2 consecutive additions of $CaCl_2$ (20 µM), followed by FCCP (4 µM). The difference between the fluorescence at complete depolarization by FCCP ($F_u$) and that just prior to the first $CaCl_2$ addition ($F_{GM}$) was presented as the extent of MMP.

## Analyses of mitochondrial calcium retention capacity (mCRC) and mitochondrial Ca²⁺ uptake

Mitochondria (500 µg) in RB were energized by adding 5 mM glutamate and 2.5 mM malate[73]. Thapsigargin (1 µM) was added to inhibit the $Ca^{2+}$ uptake by the ER that often contaminates the mitochondrial fraction. The $Ca^{2+}$ indicator arsenazo III (100 µM) was added, and kinetic measurements of absorbance was performed at 650 nm with a reference at 685 nm (BioMate 3 S, Thermo Scientific). After one minute, 40 µM $CaCl_2$ was added at every 2 minutes until no further $Ca^{2+}$ uptake was observed. Time to MPT was presented as the extent of mCRC. The same assay conditions were used for the mitochondrial $Ca^{2+}$ uptake assay. The $Ca^{2+}$ uptake rate was calculated by the slope of the $Ca^{2+}$ decrease.

## Mitochondrial swelling assays

$Ca^{2+}$-induced mitochondrial swelling was measured by 90° light scattering at 540 nm (APD130A2 Avalanche Photodetector, Thorlabs Inc.). $CaCl_2$ (200 µM) was added to the mitochondria (250 µg) energized with 5 mM glutamate and 2.5 mM malate in the presence and absence of CsA (1 µM), and swelling was monitored. After 8 minutes, ala-methicin (2 µM) was added to induce maximum swelling. The extent of swelling was presented as % of the max swelling.

## Analyses of mitochondrial ROS production

Mitochondrial ROS production was analyzed by using the Amplex Red $H_2O_2$ assay kit (Invitrogen) per manufacturer's instruction. Mitochondria (250 µg) was added to the reaction mixture containing Amplex Red (50 µM), horse radish peroxidase (0.1 U/ml), superoxide dismutase (80 U/ml), and 5 mM glutamate / 2.5 mM malate or 5 mM succinate, and the fluorescence was measured at 590 nm with 540 nm excitation (APD130A2 Avalanche Photodetector, Thorlabs Inc.).

## Statistical analyses

Proteomics data were analyzed as described above. All other statistical analyses were performed with the GraphPad Prism 9 software. Quantitative data were presented as means with SD unless specified otherwise. Differences between groups were evaluated for statistical significance with t test or ANOVA with multiple comparisons test. $P < 0.05$ was considered a statistically significant difference.

## Reporting summary

Further information on research design is available in the Nature Portfolio Reporting Summary linked to this article.

## Data availability

All data generated or analyzed during this study are included in this published article (and its supplementary information files). Source data for figures are provided with the paper. The mass spectrometry proteomics data have been deposited to the ProteomeXchange Consortium via the PRIDE partner repository with the dataset identifier PXD040556. Source data are provided with this paper.

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

## Acknowledgements

We thank Roxan Ara and Asamoa Bosomtwi in the Small Animal Imaging Shared Resource, and Donna Kumiski and Tania Green in the Histology Core at Augusta University. We also thank Kelsea Cristillo in the electron microscopy resource in the Center for Advanced Research Technologies at the University of Rochester School of Medicine. This work was supported by NIH HL093671, EY031483 and DK136753 to YY, GM144103 to HS, DE028351 and DE032084 to YT.

## Author contributions
H.L. and Y.Y. conceived the project and designed the study; H.L., Y.Y., T.J.L., C.A.G., W.Z., and W.X. performed experiments; K.L.B. and A.S. assisted with experiments; H.S. provided mice; H.L., Y.Y., Y.T., and H.S. discussed data; H.L., Y.Y., T.J.L., C.A.G., W.X., and W.Z. contributed to writing and provided feedback.

## Competing interests
The authors declare no competing interests.
