## [Peer Review File · Nature Communications]

The mitochondrial fusion protein OPA1 is dispensable in the liver and its absence induces mitohormesis to protect liver from drug-induced injuryREVIEWER COMMENTS

Reviewer #1 (Remarks to the Author):

The authors characterized an OPA-1 liver specific knockout mouse and assessed its susceptibility to acetaminophen hepatotoxicity. Deficiency of OPA-1 triggered a number of adaptive responses, which seem to be responsible for maintaining normal liver function and assured the health of the animal. Cyp2E1 levels were dramatically reduced.

1. The authors carefully characterized morphological and biochemical changes in liver-specific OPA-1 KO mice. The observations are interesting but descriptive.
2. One observation that seem to be surprising at first glance is the fact that these OPA-1 deficient mice are resistant to an APAP overdose. Given the potential impairment of mitochondrial function in these animals, it would have been expected that a toxin that works through mitochondrial dysfunction would generate a higher injury. However, the fact that JNK activation was dramatically reduced (Figure 5E) clearly indicates that the block had to be upstream of JNK activation. Since JNK is activated through binding of the reactive metabolite NAPQI to mitochondrial protein, there had to be less NAPQI formation in the OPA-1 KO mice. Indeed, the authors observed that the cyp2E1 protein levels in OPA-1 KO mice were reduced by 70% (Figure 8A,B), which completely explains the lack of injury in these animals and would explain all the beneficial effects shown in Figure 5-6. The only further confirmation of these results would be to measure reduced APAP protein adducts and reduced (Cyp2E1-dependent) oxidative metabolites of APAP, e.g. APAP-GSH, APAP-Cysteine and APAP-NAC, measured within 1 h after APAP administration. Since all other effects are downstream of the reactive metabolite formation, it is unclear which other adaptive effects reported in Figure 7-8 may also contribute to the protection. However, there can be no question that cyp2e1 downregulation is the main cause of the protection.

Reviewer #2 (Remarks to the Author):

The authors of this study conducted an experiment involving the deletion of OPA1, a protein that serves a dual function in mediating mitochondrial inner membrane fusion and cristae maintenance, in the livers of adult mice. Remarkably, the resultant mice displayed no negative effects on their phenotype, despite the disruption of mitochondrial morphology. Further analyses of the liver mitochondrial function indicated that the stress response elicited by the deletion of OPA1 led to the preservation of mitochondrial function, which is commonly referred to as mitohormesis. Although the manuscript is well-written and the results are intriguing, there are still several questions that require clarification.

1. Why do OPA LKO mice do not weight gain despite consuming the same diet as wild-type mice and showing no differences in body fat mass? It is essential to examine metabolic parameters, including respiratory quotient, at the individual mouse level. Can FGF21 be found in the serum, despite its increase in the liver?

2. The authors draw the conclusion that hepatic OPA1 is dispensable for survival owing to the incidence of mitohormesis. Nonetheless, this conclusion is based on a 14-week study and necessitates long-term monitoring. Moreover, what would occur if a metabolic burden, such as a high-fat diet, was applied instead of an acute liver injury model caused by APAP?

3. The authors mention that an increase in ALT does not imply liver damage, but what about other markers such as LDH and AST?

4. Are there differences in the amount of mitochondrial DNA and ATP content in the liver of OPA LKO and wild-type mice?

Minor comments

1. Please add a highly magnified photo in Fig.1 F

2. The arrow in Fig.2E is not explained.

3. Figs. 2H, 2I 2J, and 5E bar charts do not have individual data.

4. Please provide quantitative analysis of the western blots in Figs 1H, 4C-G.

5. There is no information on the equipment used for experiments such as MMP evaluation, mCRC analysis, mitochondrial swelling assay, and ROS measurements.

Reviewer #3 (Remarks to the Author):

In this manuscript, Lee et al. analyse the consequences of ablating OPA1 in the liver and conclude that, although some alterations are observed, overall mitochondrial functionality is not affected in the OPA1-KO mouse livers. In addition, they observe that the absence of OPA1 protects the liver from drug-induced toxicity.

The results are clearly presented and all the observations are well sustained by the experimental evidence shown throughout the manuscript.

However, the interpretation of two key points of the paper might not be completely accurate.

1) The authors focus their attention on the alterations in Complex V (CV) assembly state observed by Blue-Native gel electrophoresis (Figure 4G) and conclude that OPA1 must be involved on CV assembly somehow. However, they show that also Complexes I and IV are low in the OPA1-KO liver mitochondria (Figure 4C and 4F), with an accumulation of a CIV subassembly (Figure 4F). Also Complex II seems to be increased in the OPA1-KO samples (Figure 4E). Taken together, these alterations are compatible with defects in the mitochondrial DNA (mtDNA). Patient-derived cells with mutations in MT-ATP6, and even rho0 cells (cells devoid of mtDNA) accumulate subassemblies of the F1 fraction of CV. OPA1 has been suggested to be involved in the maintenance of mtDNA in some reports (for example PMID: 20974897), so it is plausible that the OXPHOS complex alterations observed in the OPA1-KO livers stem from defects in mtDNA, such as depletion and/or deletions. The authors should analyse mtDNA levels and integrity in the OPA1-KO livers.

On a separate note: These mice have a mixed C57BL/6 and 129Sv background and this is noticeable because in some of the individual samples, the band that corresponds to respiratory chain supercomplex III₂+IV is present and in some it is absent (Figure 4D). This is because the C57BL/6 genome encodes a non-functional allele of Cox7a2l, also known as Scaf1 (PMID: 23812712). There is a controversy as to whether the absence of this protein impacts mitochondrial function and physiological fitness in the mouse (PMID: 25470551 and PMID: 32637615). Can the authors see any differences in any of the analysed parameters between the samples that contain supercomplex III₂+IV and those in which it is absent?

2) The authors clearly show the hallmarks of the integrated stress response in the OPA1-KO livers. They also show that mitochondrial permeability and mitochondrial functionality are preserved after treatment with APAP in the OPA1-KO livers. In view of these evidences they categorically state that the protection from APAP toxicity is caused by a mitohormetic response. While the evidences point out to this, it is not demonstrated directly. The way to demonstrate it would be using a genetic or a pharmacological model inhibiting the stress responses and then checking whether the livers are not protected from APAP toxicity any more. Understandably, this is a very difficult request to fulfil at this point. Therefore, I recommend to describe this finding in a more hypothetical way along the manuscript and propose the experiments that could follow to unequivocally determine the causality of the observed phenomenon.

NCOMMS-23-06798

Title: The mitochondrial fusion protein OPA1 is dispensable in the liver and its absence induces mitohormesis to protect liver from drug-induced injury

We thank all the reviewers for their valuable input and helpful suggestions. We have taken these concerns seriously and revised the manuscript accordingly. Here we present point-by-point responses to reviewers' comments. More detailed accounts in context can be found in the main manuscript. The revised parts are marked as red fonts in the accompanied document.

REVIEWER COMMENTS

Reviewer #1 (Remarks to the Author):

The authors characterized an OPA-1 liver specific knockout mouse and assessed its susceptibility to acetaminophen hepatotoxicity. Deficiency of OPA-1 triggered a number of adaptive responses, which seem to be responsible for maintaining normal liver function and assured the health of the animal. Cyp2E1 levels were dramatically reduced.

1. The authors carefully characterized morphological and biochemical changes in liver-specific OPA-1 KO mice. The observations are interesting but descriptive.
2. One observation that seem to be surprising at first glance is the fact that these OPA-1 deficient mice are resistant to an APAP overdose. Given the potential impairment of mitochondrial function in these animals, it would have been expected that a toxin that works through mitochondrial dysfunction would generate a higher injury. However, the fact that JNK activation was dramatically reduced (Figure 5E) clearly indicates that the block had to be upstream of JNK activation. Since JNK is activated through binding of the reactive metabolite NAPQI to mitochondrial protein, there had to be less NAPQI formation in the OPA-1 KO mice. Indeed, the authors observed that the cyp2E1 protein levels in OPA-1 KO mice were reduced by 70% (Figure 8A,B), which completely explains the lack of injury in these animals and would explain all the beneficial effects shown in Figure 5-6. The only further confirmation of these results would be to measure reduced APAP protein adducts and reduced (Cyp2E1-dependent) oxidative metabolites of APAP, e.g. APAP-GSH, APAP-Cysteine and APAP-NAC, measured within 1 h after APAP administration. Since all other effects are downstream of the reactive metabolite formation, it is unclear which other adaptive effects reported in Figure 7-8 may also contribute to the protection. However, there can be no question that cyp2e1 downregulation is the main cause of the protection.

Response to Reviewer #1

1. *The authors carefully characterized morphological and biochemical changes in liver-specific OPA-1 KO mice. The observations are interesting but descriptive.*

- While the manuscript might be descriptive due to new and unexpected findings, we believe that the newly added data in revision have added more mechanistic information.

2. One observation that seem to be surprising at first glance is the fact that these OPA-1 deficient mice are resistant to an APAP overdose. Given the potential impairment of mitochondrial function in these animals, it would have been expected that a toxin that works through mitochondrial dysfunction would generate a higher injury. However, the fact that JNK activation was dramatically reduced (Figure 5E) clearly indicates that the block had to be upstream of JNK activation. Since JNK is activated through binding of the reactive metabolite NAPQI to mitochondrial protein, there had to be less NAPQI formation in the OPA-1 KO mice. Indeed, the authors observed that the cyp2E1 protein levels in OPA-1 KO mice were reduced by 70% (Figure 8A,B), which completely explains the lack of injury in these animals and would explain all the beneficial effects shown in Figure 5-6. The only further confirmation of these results would be to measure reduced APAP protein adducts and reduced (Cyp2E1-dependent) oxidative metabolites of APAP, e.g. APAP-GSH, APAP-Cysteine and APAP-NAC, measured within 1 h after APAP administration. Since all other effects are downstream of the reactive metabolite formation, it is unclear which other adaptive effects reported in Figure 7-8 may also contribute to the protection. However, there can be no question that cyp2e1 downregulation is the main cause of the protection.

- We appreciate this comment. We too were surprised by the lack of liver injury in OPA1-KO liver, which prompted us further investigation. As the reviewer pointed out, it is possible that a decrease in CYP2E1 in OPA1-KO liver may be sufficient to provide the complete protection. However, we observed a significant decrease in GSH in APAP-treated OPA1-KO mice, suggesting that a substantial amount of NAPQI was still produced and formed the APAP-cysteine adduct. Per reviewer's suggestion, we measured the APAP-CYS level and found that OPA1 KO decreased it, but a considerable amount was still present, consistent with the GSH data (**Figure 7h**). The level of APAP-CYS formed in OPA1-KO liver was similar to that produced by a lower dose of APAP that caused a moderate liver injury (PMID 23571099). In contrast, no liver injury was observed consistently in OPA1-KO liver with this level of APAP-CYS, suggesting the presence of additional protection mechanisms. Our data indicate that, in addition to an early protection by decreased CYP2E1, OPA1 LKO provides a protection by mitochondrial reinforcement. This new information is included in the Results and Discussion.

Reviewer #2 (Remarks to the Author):

The authors of this study conducted an experiment involving the deletion of OPA1, a protein that serves a dual function in mediating mitochondrial inner membrane fusion and cristae maintenance, in the livers of adult mice. Remarkably, the resultant mice displayed no negative effects on their phenotype, despite the disruption of mitochondrial morphology. Further analyses of the liver mitochondrial function indicated that the stress response elicited by the deletion of OPA1 led to the preservation of mitochondrial function, which is commonly referred to as mitohormesis. Although the manuscript is well-written and the results are intriguing, there are still several questions that require clarification.

1. Why do OPA LKO mice do not weight gain despite consuming the same diet as wild-type mice and showing no differences in body fat mass? It is essential to examine metabolic parameters, including respiratory quotient, at the individual mouse level. Can FGF21 be found in the serum, despite its increase in the liver?
2. The authors draw the conclusion that hepatic OPA1 is dispensable for survival owing to the incidence of mitohormesis. Nonetheless, this conclusion is based on a 14-week study and necessitates long-term monitoring. Moreover, what would occur if a metabolic burden, such as a high-fat diet, was applied instead of an acute liver injury model caused by APAP?
3. The authors mention that an increase in ALT does not imply liver damage, but what about other markers such as LDH and AST?
4. Are there differences in the amount of mitochondrial DNA and ATP content in the liver of OPA LKO and wild-type mice?

Minor comments

1. Please add a highly magnified photo in Fig.1 F
2. The arrow in Fig.2E is not explained.
3. Figs. 2H, 2I 2J, and 5E bar charts do not have individual data.
4. Please provide quantitative analysis of the western blots in Figs 1H, 4C-G.
5. There is no information on the equipment used for experiments such as MMP evaluation, mCRC analysis, mitochondrial swelling assay, and ROS measurements.

Response to Reviewer #2

1. Why do OPA LKO mice do not weight gain despite consuming the same diet as wild-type mice and showing no differences in body fat mass? It is essential to examine metabolic parameters, including respiratory quotient, at the individual mouse level. Can FGF21 be found in the serum, despite its increase in the liver?

- We measured metabolic parameters, and the data are included in **Figure 1**. OPA1-LKO mice showed increased VO_2 , VCO_2 , and respiratory exchange ratio as well as a changed feeding behavior. While the exact mechanisms of these changes are unclear at this point, it is interesting that OPA1 KO in the liver resulted in no harmful effect, but induces a robust whole body metabolic adaptation.
- We found an increase of serum FGF21 (**suppl. Fig. S2c**), as the circulating FGF21 is mainly derived from the liver. It was shown previously that systemic administration of FGF21 ameliorates insulin resistance and obesity. Thus, it is possible that smaller body size, improved glucose tolerance, and altered metabolic behavior of OPA1-LKO mice may be in part the systemic effect of the increased level of circulating FGF21.

2. The authors draw the conclusion that hepatic OPA1 is dispensable for survival owing to the incidence of mitohormesis. Nonetheless, this conclusion is based on a 14-week study and necessitates long-term monitoring.

- This is a valid point. However, our long-term monitoring of mice following AAV-Cre administration showed the increases of OPA1 expression and body weight after 16 weeks, indicating that AAV-mediated Cre expression becomes ineffective after 16 weeks. This information was included in the Discussion and in **suppl. Fig. S7a and b**. While this limits the study for the long-term effect of OPA1 LKO, 8 – 12 weeks KO is likely sufficiently long to test the OPA1-KO effect. For example, acute muscle OPA1-KO mice die in 8 – 12 weeks.

Moreover, what would occur if a metabolic burden, such as a high-fat diet, was applied instead of an acute liver injury model caused by APAP?

- We tested the effect of OPA1 LKO on metabolic burden by 12-week HFD. We found that OPA1 LKO completely prevented obesity and greatly decreased hepatic steatosis (**suppl. Fig. S6**). These data indicate that the mitohormetic effect induced by OPA1 LKO is likely broad, encompassing DILI, metabolic burden, and possibly other liver pathologies.

3. The authors mention that an increase in ALT does not imply liver damage, but what about other markers such as LDH and AST?

- We found that, similar to ALT, the serum AST level was also increased in OPA1-LKO mice, which was still within the reference range (**Fig. 2d**). However, LDH was not different (**Fig. 2e**).

4. Are there differences in the amount of mitochondrial DNA and ATP content in the liver of OPA LKO and wild-type mice?

- Liver ATP level was measured and there was no difference in floxed and OPA1-LKO mice (**Fig. 3k**).
- We also tested the mtDNA level and found that the level of mtDNA in OPA1 KO liver was decreased to an approximately half of the control liver (**Fig. 6c**). Please see the response to reviewer 3 below.

Minor comments

1. *Please add a highly magnified photo in Fig.1 F* – Additional images were provided in **Fig. 2a**.
2. *The arrow in Fig.2E is not explained* – This information was added in the text.
3. *Figs. 2H, 2I 2J, and 5E bar charts do not have individual data* – The graphs were fixed.
4. *Please provide quantitative analysis of the western blots in Figs 1H, 4C-G* – Quantifications were added.
5. *There is no information on the equipment used for experiments such as MMP evaluation, mCRC analysis, mitochondrial swelling assay, and ROS measurements* – This information was added.

Reviewer #3 (Remarks to the Author):

In this manuscript, Lee et al. analyse the consequences of ablating OPA1 in the liver and conclude that, although some alterations are observed, overall mitochondrial functionality is not affected in the OPA1-KO mouse livers. In addition, they observe that the absence of OPA1 protects the liver from drug-induced toxicity.

The results are clearly presented and all the observations are well sustained by the experimental evidence shown throughout the manuscript.

However, the interpretation of two key points of the paper might not be completely accurate.

1) The authors focus their attention on the alterations in Complex V (CV) assembly state observed by Blue-Native gel electrophoresis (Figure 4G) and conclude that OPA1 must be involved on CV assembly somehow. However, they show that also Complexes I and IV are low in the OPA1-KO liver mitochondria (Figure 4C and 4F), with an accumulation of a CIV subassembly (Figure 4F). Also Complex II seems to be increased in the OPA1-KO samples (Figure 4E). Taken together, these alterations are compatible with defects in the mitochondrial DNA (mtDNA).

Patient-derived cells with mutations in MT-ATP6, and even rho0 cells (cells devoid of mtDNA) accumulate subassemblies of the F1 fraction of CV. OPA1 has been suggested to be involved in the maintenance of mtDNA in some reports (for example PMID: 20974897), so it is plausible that the OXPHOS complex alterations observed in the OPA1-KO livers stem from defects in mtDNA, such as depletion and/or deletions. The authors should analyse mtDNA levels and integrity in the OPA1-KO livers.

On a separate note: These mice have a mixed C57BL/6 and 129Sv background and this is noticeable because in some of the individual samples, the band that corresponds to respiratory chain supercomplex III₂+IV is present and in some it is absent (Figure 4D). This is because the C57BL/6 genome encodes a non-functional allele of Cox7a2l, also known as Scaf1 (PMID: 23812712). There is a controversy as to whether the absence of this protein impacts mitochondrial function and physiological fitness in the mouse (PMID: 25470551 and PMID: 32637615). Can the authors see any differences in any of the analysed parameters between the samples that contain supercomplex III₂+IV and those in which it is absent?

2) The authors clearly show the hallmarks of the integrated stress response in the OPA1-KO livers. They also show that mitochondrial permeability and mitochondrial functionality are preserved after treatment with APAP in the OPA1-KO livers. In view of these evidences they categorically state that the protection from APAP toxicity is caused by a mitohormetic response. While the evidences point out to this, it is not demonstrated directly. The way to demonstrate it would be using a genetic or a pharmacological model inhibiting the stress responses and then checking whether the livers are not protected from APAP toxicity any more. Understandably, this is a very difficult request to fulfil at this point. Therefore, I recommend to describe this finding in a more hypothetical way along the manuscript and propose the experiments that could follow to unequivocally determine the causality of the observed phenomenon.

Response to Reviewer #3

1. The authors focus their attention on the alterations in Complex V (CV) assembly state observed by Blue-Native gel electrophoresis (Figure 4G) and conclude that OPA1 must be involved on CV assembly somehow. However, they show that also Complexes I and IV are low in the OPA1-KO liver mitochondria (Figure 4C and 4F), with an accumulation of a CIV subassembly (Figure 4F). Also Complex II seems to be increased in the OPA1-KO samples (Figure 4E). Taken together, these alterations are compatible with defects in the mitochondrial DNA (mtDNA). Patient-derived cells with mutations in MT-ATP6, and even rho0 cells (cells devoid of mtDNA) accumulate subassemblies of the F1 fraction of CV. OPA1 has been suggested to be involved in the maintenance of mtDNA in some reports (for example PMID: 20974897), so it is plausible that the OXPHOS complex alterations observed in the OPA1-KO livers stem from defects in mtDNA, such as depletion and/or deletions. The authors should analyse mtDNA levels and integrity in the OPA1-KO livers.

- We added that there was a small accumulation of a complex IV subassembly.

- The quantification of mtDNA showed that OPA1 KO decreased the mtDNA content by approximately a half (**Fig. 6c**). We further tested whether the complex V assembly defect is a specific effect of OPA1 KO or a general phenomenon from a lack of mtDNA by using OPA1-KO and Mfn-DKO MEF cell lines that showed similarly decreased mtDNA levels (**Fig. 6d – g**). Our data indicate that the complex V assembly defect in OPA1-KO liver is a specific effect of OPA1 KO. In the liver, OPA1 KO induces an efficient ISR, which increases mitochondrial biogenesis (including mtDNA) to prevent a large loss of mtDNA and thus maintain mitochondrial function. In the case of mitochondrial heteroplasmy, it was shown that more than 70% of mutated mtDNA is required to manifest functional defect. Pronounced complex V assembly defect by OPA1 KO with a functionally inconsequential reduction in mtDNA level suggests a direct and specific role of the OPA1 molecule in the assembly or stability of complex V. These new data and our reasoning are included in the revised manuscript.

2. On a separate note: These mice have a mixed C57BL/6 and 129Sv background and this is noticeable because in some of the individual samples, the band that corresponds to respiratory chain supercomplex III₂+IV is present and in some it is absent (Figure 4D). This is because the C57BL/6 genome encodes a non-functional allele of Cox7a2l, also known as Scaf1 (PMID: 23812712). There is a controversy as to whether the absence of this protein impacts mitochondrial function and physiological fitness in the mouse (PMID: 25470551 and PMID: 32637615). Can the authors see any differences in any of the analysed parameters between the samples that contain supercomplex III₂+IV and those in which it is absent?

- We examined the SCAF1 alleles in our mice and confirmed its role in the formation of [III₂ + IV] (**suppl. Fig. S3c, and d**). We have not observed any correlation between SCAF1 and functional aspect of mitochondria. Irrespective of the SCAF1 allele, OPA1-LKO mice consistently showed halted weight gain, ISR induction, and changes in respiratory complexes.

3. The authors clearly show the hallmarks of the integrated stress response in the OPA1-KO livers. They also show that mitochondrial permeability and mitochondrial functionality are preserved after treatment with APAP in the OPA1-KO livers. In view of these evidences they categorically state that the protection from APAP toxicity is caused by a mitohormetic response. While the evidences point out to this, it is not demonstrated directly. The way to demonstrate it would be using a genetic or a pharmacological model inhibiting the stress responses and then checking whether the livers are not protected from APAP toxicity any more. Understandably, this is a very difficult request to fulfil at this point. Therefore, I recommend to describe this finding in a more hypothetical way along the manuscript and propose the experiments that could follow to unequivocally determine the causality of the observed phenomenon.

- We included following sentences in the Discussion: “While our data point to the ISR being the major factor contributing to the mitohormetic effect of OPA1 LKO, more direct evidence would be necessary to confirm this notion. FGF21 is a major ISR effector, and

FGF21 KO was previously shown to abolish the metabolic benefit induced by liver ISR in HFD. It will be interesting to test if FGF21 KO eliminates the protective effect of OPA1 LKO in APAP toxicity. For directly test the role of ISR, the chemical inhibitor of ISR, ISRIB can also be considered.”

REVIEWERS' COMMENTS

Reviewer #1 (Remarks to the Author):

The authors satisfactorily addressed the reviewers' comments.

Reviewer #2 (Remarks to the Author):

The authors meticulously addressed the reviewer's feedback with utmost precision, resulting in a significant enhancement of the manuscript.

Reviewer #3 (Remarks to the Author):

The authors have addressed the reviewers' points of concern satisfactorily for the most part, making the results and claims of this manuscript much more solid. However, even if this is not the main point of this article, I still disagree with the authors' interpretation that the complex V defect is specifically due to the lack of OPA1 in the liver. In figure 5 they show a decrease in complex I and complex IV levels, which is typically observed in the cases of mtDNA depletion or defects in the expression of mitochondrial DNA. They show also a profound depletion of mtDNA in the OPA1-KO livers and MEFs, as well as the accumulation of partially assembled F1 particle of complex V also in the Mfn-KO MEFs with a similar level of mtDNA depletion.

REVIEWERS' COMMENTS

Reviewer #1 (Remarks to the Author):

The authors satisfactorily addressed the reviewers' comments.

Reviewer #2 (Remarks to the Author):

The authors meticulously addressed the reviewer's feedback with utmost precision, resulting in a significant enhancement of the manuscript.

Reviewer #3 (Remarks to the Author):

The authors have addressed the reviewers' points of concern satisfactorily for the most part, making the results and claims of this manuscript much more solid. However, even if this is not the main point of this article, I still disagree with the authors' interpretation that the complex V defect is specifically due to the lack of OPA1 in the liver. In figure 5 they show a decrease in complex I and complex IV levels, which is typically observed in the cases of mtDNA depletion or defects in the expression of mitochondrial DNA. They show also a profound depletion of mtDNA in the OPA1-KO livers and MEFs, as well as the accumulation of partially assembled F1 particle of complex V also in the Mfn-KO MEFs with a similar level of mtDNA depletion.

Response to Reviewer #3

1. The authors have addressed the reviewers' points of concern satisfactorily for the most part, making the results and claims of this manuscript much more solid. However, even if this is not the main point of this article, I still disagree with the authors' interpretation that the complex V defect is specifically due to the lack of OPA1 in the liver. In figure 5 they show a decrease in complex I and complex IV levels, which is typically observed in the cases of mtDNA depletion or defects in the expression of mitochondrial DNA. They show also a profound depletion of mtDNA in the OPA1-KO livers and MEFs, as well as the accumulation of partially assembled F1 particle of complex V also in the Mfn-KO MEFs with a similar level of mtDNA depletion.

- We toned down our interpretation by saying “potentially specific role”. In addition, we added following sentences. “Although our MEFs data suggest a specific role of OPA1 in complex V assembly, to what extent a decrease in mtDNA content in OPA1 KO liver contributes to the complex V assembly defect needs to be further investigated. While the decrease in complexes I and IV in OPA1-KO liver is suggestive of the depleted mtDNA phenotype to some degree, we observed the increase in complexes II and V in OPA1 KO liver, suggesting an involvement of stress response in the contents of the respiratory complexes.”